# New Psychoactive Substances Intoxications and Fatalities during the COVID-19 Epidemic

**DOI:** 10.3390/biology12020273

**Published:** 2023-02-08

**Authors:** Alfredo Fabrizio Lo Faro, Diletta Berardinelli, Tommaso Cassano, Gregory Dendramis, Eva Montanari, Angelo Montana, Paolo Berretta, Simona Zaami, Francesco Paolo Busardò, Marilyn Ann Huestis

**Affiliations:** 1Department of Excellence Biomedical Sciences and Public Health, Marche Polytechnic University, 60121 Ancona, Italy; 2Department of Medical and Surgical Sciences, University of Foggia, Via Luigi Pinto, c/o Policlinico “Riuniti” di Foggia, 71122 Foggia, Italy; 3Department of Cardiology, ARNAS Ospedale Civico di Cristina Benfratelli, 90127 Palermo, Italy; 4National Centre on Addiction and Doping, Istituto Superiore di Sanità, 00161 Rome, Italy; 5Department of Anatomical, Histological, Forensic and Orthopedic Sciences, “Sapienza” University of Rome, 00161 Rome, Italy; 6Institute of Emerging Health Professions, Thomas Jefferson University, Philadelphia, PA 19107, USA

**Keywords:** COVID-19, new psychoactive substances, fatalities, intoxications

## Abstract

**Simple Summary:**

We presented a comprehensive, systematic literature review of all published New Psychoactive Substances (NPS)-related intoxications and fatalities during the COVID-19 pandemic (from January 2020 to March 2022). Public implications, such as isolation and social distancing, may have reduced consumption of some drugs. These stressful conditions brought an increase in the use of other drugs, with the illicit market and related misuse of drugs moving to different drugs of abuse, such as NPS. More than 200 cases were reported in Europe, UK, USA and Japan during the pandemic period, with synthetic opioid, synthetic cannabinoids and synthetic cathinones the most representative NPS classes. Importantly, the combined consumption of several NPS classes comprised 30% of all cases. Considering that the pandemic may have reduced the capabilities of forensic toxicology laboratories to report comprehensive information, the data could have led to an underestimation.

**Abstract:**

In January 2020, the World Health Organization (WHO) issued a Public Health Emergency of International Concern, declaring the COVID-19 outbreak a pandemic in March 2020. Stringent measures decreased consumption of some drugs, moving the illicit market to alternative substances, such as New Psychoactive Substances (NPS). A systematic literature search was performed, using scientific databases such as PubMed, Scopus, Web of Science and institutional and government websites, to identify reported intoxications and fatalities from NPS during the COVID-19 pandemic. The search terms were: COVID-19, SARS-CoV-2, severe acute respiratory syndrome coronavirus 2, coronavirus disease 2019, intox*, fatal*, new psychoactive substance, novel psychoactive substance, smart drugs, new psychoactive substance, novel synthetic opioid, synthetic opioid, synthetic cathinone, bath salts, legal highs, nitazene, bath salt, legal high, synthetic cannabinoid, phenethylamine, phencyclidine, piperazine, novel benzodiazepine, benzodiazepine analogue, designer benzodiazepines, tryptamine and psychostimulant. From January 2020 to March 2022, 215 NPS exposures were reported in Europe, UK, Japan and USA. Single NPS class intoxications accounted for 25, while mixed NPS class intoxications represented only 3 cases. A total of 130 NPS single class fatalities and 56 fatalities involving mixed NPS classes were published during the pandemic. Synthetic opioids were the NPS class most abused, followed by synthetic cathinones and synthetic cannabinoids. Notably, designer benzodiazepines were frequently found in combination with fentalogues. Considering the stress to communities and healthcare systems generated by the pandemic, NPS-related information may be underestimated. However, we could not define the exact impacts of COVID-19 on processing of toxicological data, autopsy and death investigations.

## 1. Introduction

The World Health Organization (WHO) declared a Public Health Emergency of International Concern in January 2020 and characterized the outbreak as a pandemic in March 2020.

In response to the COVID-19-pandemic, lockdowns were required to prevent the spread of the disease, resulting in drastic reductions in social contact in public settings and in most individuals’ private and professional lives. Such restrictions also led to modifications in the synthesis of chemical precursors, manufacturing and distribution [1,2]. In April 2020, the European Monitoring Centre for Drugs and Drug Addiction (EMCDDA) released an alert to the EU Early Warning System Network, focusing on the potential impact of the pandemic on drug markets [1,3].

As reported in the Global Drug Survey [4,5], stressful conditions produced by pandemic restrictions brought an increase in the use of some drugs. According to this survey [4], increases in benzodiazepine use during the pandemic could have arisen from the need to manage anxiety and depression; however, benzodiazepines are highly addictive, which could have led to a serious public health problem. Stringent measures, including isolation and social distancing, resulted in lower consumption of some drugs [1], such as stimulants and hallucinogens.

The illicit market and related misuse of drugs moved to alternative substances [6], such as New Psychoactive Substances (NPS). NPS comprise a heterogenous group of substances [7], such as prescription drugs or research chemicals, which are not controlled under the 1961 Single Convention on Narcotic Drugs or the 1971 Convention on Psychotropic Substance, synthetized to mimic the psychoactive effects of commonly abused drugs. NPS consumption [8,9,10,11,12,13,14,15] can cause several adverse effects, such as acute psychosis, bradypnea, angina pectoris, migraine, headaches and life-threatening cardiovascular problems ranging from mild tachycardia to arrhythmias, myocardial infarction and even death.

Considering the many reported cardiovascular complications [16,17,18,19,20] for NPS and commonly abused drugs, and the high life-threatening risk every time an NPS enters into the market, further studies on the side effects of drug consumption, including cardiotoxicity, are required. The most recent literature on NPS did not include comprehensive systematic reviews [13,14,15] on NPS intoxications and fatalities. While synthetic opioids, synthetic cannabinoid and synthetic cathinones represented the most abused NPS classes during the COVID-19 pandemic, co-consumption of different NPS classes raised particular concern.

The pandemic overloaded the healthcare system, reducing the capability of forensic toxicology laboratories and emergency departments to detect and report NPS-related information. As a consequence, the data could be underestimated [1]. To address this concern, we aimed to identify all the intoxications and fatal cases reported, during the pandemic, caused by single or mixed NPS classes, focusing on which NPS class caused the most intoxications and/or fatalities as well as the most frequent combination of them.

Herein, we presented a comprehensive, systematic review of all published NPS-related intoxication instances and fatalities throughout the COVID-19 pandemic (from January 2020 to March 2022).

## 2. Materials and Methods

In accordance with Prisma guidelines [21], a literature search was performed to identify reported intoxications and fatalities from NPS during the COVID-19 pandemic, specifically from January 2020 to March 2022. PubMed, Scopus, Web of Science and institutional and government websites were searched by two scientists for the following terms, alone or in combination: (“COVID-19” OR “SARS-CoV-2” OR “Severe acute respiratory syndrome coronavirus 2” OR “Coronavirus disease 2019” OR intox* or fatal*) AND (“New psychoactive substance” OR “Novel psychoactive substance” OR “smart drugs” OR “Novel psychoactive substances” OR “New psychoactive substances” OR “Synthetic opioid” OR “Novel synthetic opioids” OR “Novel synthetic opioid” OR “Synthetic opioids” OR “Synthetic cathinone” OR “bath salts” OR “legal highs” OR “nitazene” OR “Synthetic cathinones” OR “bath salt” OR “legal high” OR “synthetic cannabinoid” OR “synthetic cannabinoids” OR phenethylamine OR phencyclidine OR piperazine OR “novel benzodiazepine” OR “Benzodiazepine analogue” OR “designer benzodiazepines” OR tryptamine OR Psychostimulant).

Relevant articles cited in the references of these selected publications were also considered. Articles were screened according to the following criteria:English, Italian or French language.Studies involving human exposure to NPS, alone or in combination with other drugs, from January 2020 to March 2022.Exposure confirmation through qualitative or quantitative toxicology analyses of human biological matrices.

All the data were extracted, curated and analyzed without software aid. Articles were manually screened by three of the authors. Moreover, a second check was performed to avoid researcher bias. Figure 1 shows the Prisma Flowchart of the literature search.

## 3. Results

The results of the literature search are shown in Table 1. Out of 5207 records identified, a total of 63 papers met the inclusion criteria (after duplicate removal). In Table 1, all cases reported were clustered by year. Whenever available, each case included a descriptive analysis of individual characteristics (e. g., age, gender), the class of new psychoactive substances (NPS) and relative concentrations in different biological matrices (iliac and femoral blood were reported as peripheral blood), concomitant exposure(s), date and country where the report originated.

From January 2020 to March 2022, a total of 215 NPS exposures were reported, including both single- and mixed-NPS class exposures (Figure 1). Most exposures were observed in 2020; i.e., 58% (123 cases) in 2020 [22,23,24,25,26,27,28,29,30,31,32,33,34,35,36,37,38], 32% (71 cases) in 2021 [31,32,39,40,41,42,43,44,45,46,47,48,49,50,51,52,53] and 10% (21 cases) in 2022 [47,54,55,56,57,58,59,60]. During each year from 2020–2022, the prevalence of single NPS class exposure was always higher than the prevalence of mixed NPS exposures. In 2020, 87 cases [22,23,24,25,26,27,28,29,38,61,62,63] involved a single NPS class, accounting for 68% of all consumption instances in the year (123 cases), whilst 59 (85%) and 11 (48%) single NPS class exposures were reported in 2021 [23,31,32,40,41,42,43,44,45,47,48,49,50,53,64,65] and 2022 [54,55,56,57,59,60], respectively.

### 3.1. Single NPS Class Intoxications

Considering single NPS class intoxication cases during the COVID-19 pandemic, 25 such incidents were reported [28,32,34,37,38,41,42,50,56,59,64]; 18 in 2020 [28,34,37,38], 4 in 2021 [41,42,50], and 3 from January to March 2022 [56,59,64] (Figure 2). The most common single NPS class involved in non-fatal cases was synthetic cannabinoids, with 19 reported cases. Seventeen occurred in 2020 [34,37,38], one occurred in 2021 [42] and one occurred in the period from January to March 2022 [59] (Figure 3a). Intoxication cases were for 4F-MDMB-BICA (n = 5) [38], MDMB-4en-PINACA (n = 12) [37] and 5F-ADB (n = 2) [42,59]. Median blood concentrations were not available.

Synthetic cathinones, the second most common NPS, accounted for 4 intoxications [32,50,56] (Figure 3a); 3 cases occurred in 2021 [32,56] and one [56] occurred in the first three months of 2022 (Figure 4a). Synthetic cathinones’ intoxications involved 3-MMC (n = 3) [32,50], and 1 N-ethylpentylone [56].

From January 2020 to March 2022, only two single synthetic opioid intoxications were observed (Figure 2); 1 [28] in 2020 and 1 [41] in 2021 (Figure 4a). No intoxications were reported in the first trimester of 2022. Only 1 intoxication [41] involved fentanyl consumption, detected with its metabolites. The other intoxication case [28] involved brorphine intake. As for intoxication cases related to other NPS classes [64], only one intoxication was reported for arylcyclohexylamine (4-FiBP, α-PiHP, 4-FiBF).

### 3.2. Single NPS Class Fatalities

A total of 130 fatalities [22,23,24,25,26,27,29,31,32,33,34,35,36,38,40,44,45,48,49,53,54,55,56,62,65,66,67] related to single NPS class intake were described from January 2020 to March 2022. In 2020, 68 fatalities occurred [22,23,24,25,27,29,31,32,33,34,35,36,38], with 55 deaths occurring in 2021 [23,31,32,40,44,45,48,49,53,65,67] and 7 additional fatalities in the first three months of 2022 [54,55,56,60] involving a single NPS class (Figure 2).

During the COVID-19 pandemic, the NPS class which produced the most fatalities was synthetic opioids, totaling 58 casualties [22,23,25,26,40,45,48,53,60]. In 2020, 15 fatalities [22,23,25,26] were registered after synthetic opioid consumption, with 39 deaths in 2021 [23,40,45,48,53] and 3 reported from January 2022 to March 2022 [60] (Figure 4a).

Among synthetic opioids, fentanyl was the most abused, alone and in combination with other synthetic opioids [22,23,25,40,41,45,48] (n = 43). The median fentanyl blood concentration was 18 ng/mL [23,41,45,48] (n = 36). There was 1 fatality [45] reported involving a fentanyl single consumption with its metabolites. Fentanyl intake primarily occurred with other synthetic opioids in: 18 fatal cases [23] (p-FF, acetyl fentanyl), 8 cases [25,40] (brorphine, median blood concentration = 0.7 ng/mL) 7 cases [48] (metonitazene, median blood concentration = 2.4 ng/mL), 7 cases (p-FF, detected in urine) and 1 case [26] (isotonitazene, acetyl fentanyl). Single exposures to isotonitazene (median blood concentration = 1.2 ng/mL) resulted in 4 deaths [26,68], as was true for metonitazene [48] (median blood concentration = 4.4 ng/mL).

The second most common NPS class [34,38,44,56,57] encountered was synthetic cannabinoids, with 30 deaths (Figure 3a). In total, 27 fatalities [25,38] were reported in 2020, while only 1 fatality was published in 2021 [44], and two fatal cases [56,57] occurred in the first trimester of 2022 (Figure 4a). Substances identified in fatalities were 4F-MDMB-BICA [38] (n = 21), MDMB-4en-PINACA [34,38] (n = 2) and 4F-MDMB-BINACA together with MDMB-4en-PINACA [34] (n = 3). Only one case was reported for ADB-FUBINACA [56], 5F-ADB [57], 4F-MDMB-PINACA [44] and MDMB-4en-PINACA [34] together with MMB-FUBINACA, respectively. Median blood concentrations were not available.

Synthetic cathinones, the third most abundant NPS class [24,31,32,33,35,36,49,54,55,62], accounted for 31 fatalities (Figure 3a). In the year 2020, there were 17 deaths [24,31,32,33,35,36,62] from synthetic cathinones, while 12 fatalities [31,32,49,64] were reported in 2021. In the first three months of 2022, 2 synthetic cathinone deaths [54,55] were described (Figure 4a). These fatalities involved the following substances: 3-CMC [31] (n = 9), 3-MMC [32] (n = 6), N-ethylpentylone (n = 3, median blood concentration 64.7 ng/mL) and eutylone [24] (n = 6, median blood concentration 3.1 ng/mL). Single intakes were reported for α-PVP [62], MDPHP [54], *N*-ethylhexedrone [36], 4-MEC [49], fluoro-methyl-PVP [55] and 4-MDP with 4-MEC and 4-CMC [35] together with α-PiHP.

Other fatalities were reported in 2020 from DBZPs [27] (n = 5, flualprazolam, median blood concentration 3.6 ng/mL), tryptamine [29] (n = 5, 5-Meo-DIPT, median urine concentration 2 ng/mL) and 2 from phenethylamine [43,65] (2-MAPB and 3-MeO-PCP) consumption in 2021 (Table 1). Eighty-five percent of the single NPS exposures were detected in males.

### 3.3. Mixed NPS Class Intoxications

During the COVID-19 pandemic, only 3 intoxications [51,52,60] involved more than one NPS class: 2 [52,60] in 2021 and 1 [51] from January to March 2022 (Figure 3b). In one NPS intoxication [52], synthetic cathinones were combined with arylcyclohexylamine. The last 2 intoxications [51,60] involved more than 2 NPS classes, including synthetic opioids, synthetic cathinones, natural NPS and DBZP for 1 case [60], and arylcyclohexylamines, synthetic cathinones, DBZP and natural NPS [51] for another case (Figure 4b).

### 3.4. Mixed NPS Class Fatalities

From January 2020 to March 2022, 56 fatalities [23,27,30,39,48,58,60,68] involving mixed NPS classes were recorded (Figure 3b). The highest numbers of multiple NPS class exposures were observed in 2020 [24,25,26,27,30,34], with all 36 resulting in death. There were 10 reports [23,39,48,68] of mixed NPS intake in 2021 and 10 [58,60] in 2022 (Figure 4b).

Synthetic opioids and DZBP constituted the most common and deadly drug combination, with all 45 cases resulting in death. The primary DZBP found in these fatal coexposures was flualprazolam mixed with isotonitazene [25,27] (n = 8), with median blood concentrations of 6.2 ng/mL flualprazolam and 1.7 ng/mL isotonitazene. The other common flualprazolam combination was with fentanyl and brorphine [25] (n = 8). Where provided, the median blood and/or urine concentrations were 3.6 ng/mL flualprazolam and 18.5 ng/mL fentanyl. Brorphine median blood concentration was 1.1 ng/mL.

Similar to flualprazolam, etizolam (median blood concentration 12.5 ng/mL) was reported in 5 cases [26], in tandem with isotonitazene (median blood concentration 2.1 ng/mL, n = 4) or metonitazene [48] (n = 1). Etizolam and isotonitazene were also combined with fentanyl (n = 2) or flualprazolam [26] (n = 1). One case involved four different NPS [60] (i.e., etizolam, 2-methyl-AP-237 2-FDCK and 3-OH-PCP. Other synthetic opioids and DBZP cases are described in Table 1. Out of 46 fatal cases, 31 [25,26,27] occurred in 2020, 7 [39,48] in 2021 and 7 [60,69] in 2022.

There were three cases involving synthetic opioids and synthetic cathinones, with 2 [23,24] reporting fentanyl (median blood concentration 15.1 ng/mL) and eutylone (median blood concentration 69 ng/mL). Another combination case [48] identified metonitazene, butonitazene and *N*-ethylpentylone (NEP). A single case reported consumption of eutylone and synthetic cannabinoids (Figure 4b).

There was a single fatal tryptamine and arylcyclohexylamine case [68] and a single fatal synthetic cathinone and arylcyclohexylamine case [52] described. Considering cases with more than two NPS classes, 3 fatalities [26,51,60] were identified, including coexposures of synthetic opioids, DBZP, arylcyclohexylamines and piperazines (Figure 4b). Mixed NPS cases occurred more frequently in males (56%).

### 3.5. Analytical Methods

The analytical technique most encountered, where reported, (n = 117), in the cases shown in Table 1, was liquid chromatography high resolution mass spectrometry (LC-HRMS/MS, HPLC-HRMS/MS, LC-QTOF) [24,25,26,27,28,29,31,34,37,40,42,43,44,45,48,49,50,52,53,54,55,58,59,61,62,63,64,65,68], followed by liquid chromatography mass spectrometry (LC-MS/MS, HPLC-MS/MS, UHPLC-MS/MS) (n = 69) and gas chromatography mass spectrometry (GC-MS, GC-MS/MS, HS-GC-MS) (n = 21) [22,29,35,43,44,47,49,50,55,56,59,62,68]. Other studies also used LC-DAD, HPLC-DAD, HS-GC-FID and NMR [35,50,56,59,68].

**Table 1 biology-12-00273-t001:** NPS-related fatalities and intoxications reported in the literature during COVID-19 (January 2020–March 2022).

Sex, Age, Type	NPS	Toxicology (ng/mL)	Coexposure (ng/mL)	Date, Country	Ref.
M 48 Fatal	MDPHP	BP 399, U 222, G 50	Ethanol B 9.1 mM/L	2022, Italy	[54]
M 30 Fatal	Fluoro-methyl-PVP	BP 26, BC 30, VH 20		2022, USA	[55]
M 29 Intox	4-FiBP α-PiHP 4-FiBF	BP 87.7, U 2291 BP 5.0, U 722 B 119, U 289, VH 101, Brain 112 ng/g, L 1540 ng/g	Amphetamine B +	2022, Poland	[64]
M 41 Fatal	ADB-FUBINACA	BP 105, BC 204, U 13, B 320, G 4640, CSF 65, K 41 ng/g, L 45 ng/g		2022, Hungary	[56]
M 42 Intox	*N*-ethylpentylone	BP 1.5, BC 1.7, U 137, B 170, G 213, K 1.5 ng/g		2022, Hungary	[56]
M 16 Fatal	5-F-ADB metabolite	BP 79.8	Haloperidol B 5.4, Cotinine B +, Caffeine B +	2022, USA	[57]
M 55 Fatal	Methoxiphenidine 3-MMC	BP 660 μg/L, BC 254 μg/L, H 13 ng/g, U 238 μg/L		2022, France	[70]
M 31 Intox	5F-ADB 5F-ADB *N*-(5-OH-pentyl)	P + P +	THC P 8.0 μg/L, OH-THC P 4.0 μg/L, THC-COOH P 147 μg/L, Caffeine P +, Metamizole Metabolite P 8700 μg/L	2022, German	[59]
M 31 Intox	2-methyl AP-237 Eutylone Mitragynine 7-Hydroxymitragynine Etizolam	B 21 Matrix not indicated + Matrix not indicated + Matrix not indicated + Matrix not indicated +	Matrix not indicated Caffeine, O-desmethyltramadol, 7-aminoclonazepam, Methadone, Naloxone, Pyrazolam, Clonazepam	2022, USA	[60]
M 29 Fatal	2-methyl AP-237	B 5800	Matrix not indicated Caffeine, Cotinine, Quinine, Naloxone, Trazodone, Phenibut B 77	2022, USA	[60]
M 35 Fatal	2-methyl AP-237 Etizolam 2-FDCK 3-OH-PCP	B 1100, U 5000, HV 720 Matrix not indicated 22 Matrix not indicated + Matrix not indicated +	Matrix not indicated Caffeine, Carisoprodol 840, Meprobamate 7300, THC 1.1, THC-COOH 6.1, Promethazine 33, Meth 45,	2022, USA	[60]
M 29 Fatal	2-methyl AP-237 Mitragynine	B 820, U 1600 Matrix not indicated 32	Matrix not indicated Caffeine, Cotinine, Naloxone, 7-aminoclonazepam 37, OH-THC 3.5, THC-COOH 57, THC 9, Tadalafil 49, Amphetamine 15	2022, USA	[60]
NA Fatal	2-methyl AP-237	B 1400	Naloxone, Trazodone, Hydroxyzine	2022, USA	[60]
M 41 Fatal	AP-238	B 87, U 120	Methadone, EDDP, THC, THC-COOH, OH-THC, Memantine, 8-aminoclonazepam	2022, USA	[60]
M 28 Fatal	AP-238 Flualprazolam	B 270, U 1200 Matrix not indicated +	Matrix not indicated Caffeine, Nicotine, Acetaminophen, THC, THC-COOH, OH-THC, 8-aminoclonazepam	2022, USA	[60]
NA 20–50 Fatal	Etonitazepyne Fentanyl DBZP	B 2.4 B + B +	Meth B +	2022, USA	[47]
NA 20–50 Fatal	Etonitazepyne Fentanyl DBZP	B 2.4 B + B +	Meth B +	2022, USA	[47]
NA 20–50 Fatal	Etonitazepyne Fentanyl DBZP	B 2.4 B + B +	Meth B +	2022, USA	[47]
NA 20–50 Fatal	Etonitazepyne Fentanyl DBZP	B 2.4 B + B +	Meth B +	2022, USA	[47]
NA 20–50 Fatal	Etonitazepyne DBZP	B + B +		2022, USA	[47]
NA 20–50 Fatal	Etonitazepyne DBZP	B + B +		2022, USA	[47]
M 31 Fatal	Fentanyl p-FF Acetyl fentanyl 4-ANPP Norfentanyl	B 49, U + U + U + U + U +	Xylazine U +, Alprazolam B 17, α-OH-alprazolam U +, THC-COOH U +	2021, USA	[23]
M 49 Fatal	Fentanyl p-FF 4-ANPP Norfentanyl	B 26, U + B 14, U + U + U +	Cocaine B 298, U +, CE B 77, U +, BE B 430, U + Volatiles B 0.19, U 0.25, HV 0.23 g/dL	2021, USA	[23]
M 33 Fatal	Fentanyl p-FF Acetyl Fentanyl 4-ANPP Norfentanyl	B 41, U + U + U + U + U +	Xylazine U +, Meth B > 100, U +, Amphetamine U +, Lidocaine U +, Gabapentine U + Volatiles B 0.031, HV 0.040 g/dL	2021, USA	[23]
M 28 Fatal	Fentanyl p-FF Acetyl Fentanyl 4-ANPP Norfentanyl	B 13, U + U + U + U + U +	Tramadol U +	2021, USA	[23]
M 45 Fatal	Fentanyl p-FF 4-ANPP Eutylone	B 4.2, U + U + U + B 28, U +	Meth B 280, U +, Amphetamine U +, THC-COOH U +	2021, USA	[23]
M 26 Fatal	Fentanyl p-FF 4-ANPP Norfentanyl	B > 2.5, U + U + U + U +	Metronizadole U +, Buspirone U +, Amphetamine > 100, Meth > 100, U +	2021, USA	[23]
M 50 Fatal	Fentanyl p-FF Acetyl Fentanyl 4-ANPP Norfentanyl	B 49, U + B 30, U + U + U + U +	Lidocaine U +, Morphine U +, 6-AM U +, BE U +	2021, USA	[23]
M 40 Fatal	Fentanyl p-FF 4-ANPP Norfentanyl	B 20, U + B 9, U + U + U +	Xylazine U +, Amphetamine < 100, U + Meth < 100, U + Cocaine < 50, U +, BE 1514, U +, THC-COOH U +, Levamisole U +, CE U +	2021, USA	[23]
M 38 Fatal	Fentanyl p-FF Acetyl Fentanyl 4-ANPP Norfentanyl	B 10, U + B 4.1, U + U + U + U +	Xylazine U +, Methadone B +, U +, Meth B > 100, U +, Methadone metabolite U +, Amphetamine U +	2021, USA	[23]
M 39 Fatal	Fentanyl p-FF Acetyl Fentanyl 4-ANPP Norfentanyl	B 3.1 U + U + U + U +	Xylazine U +, Sertraline B > 500, Desmethylsertraline B +, U +, Amphetamine B < 100, U +, Alprazolam B 14, α-OH-alprazolam U + Trazodone U +,	2021, USA	[23]
M 32 Fatal	Fentanyl p-FF Norfentanyl	B 3, U + B 3.9, U + U +		2021, USA	[23]
M 32 Fatal	Fentanyl p-FF Acetyl Fentanyl 4-ANPP Norfentanyl	B 49, U + U + U + U + U +	BE U +	2021, USA	[23]
F 58 Fatal	Fentanyl p-FF Norfentanyl	B 10, U + U + U +	Xylazine U +, Gabapentin U +, Amphetamine B > 100, U + Meth B 1.1, U +, Gabapentin U +, Chlorpheniramine U +	2021, USA	[23]
M 34 Fatal	Fentanyl p-FF Acetyl Fentanyl 4-ANPP Norfentanyl	B 32, U + B < 2.5, U + U + U + U +	Xylazine U +, Lidocaine B +, U +, Tramadol < 500, U +	2021, USA	[23]
M 27 Fatal	Fentanyl p-FF Acetyl Fentanyl 4-ANPP Norfentanyl	B 26, U + B 2.5, U + U + U + U +	Xylazine U +, Amphetamine B 150, U +, Meth B 2.6, U +, BE B 111, U +, Cocaine U +, Levamisole U +, THC-COOH U +, Morphine U +	2021, USA	[23]
F 34 Fatal	Fentanyl p-FF Acetyl Fentanyl 4-ANPP Norfentanyl	B 139, U + U + U + U + U +	Xylazine U +, Meth < 100, U +, Amphetamine U +, THC-COOH U +	2021, USA	[23]
M 34 Fatal	p-FF Norfentanyl	U + U +	Acetone B 24, Amphetamine B < 100, U +, Meth B < 100, U +	2021, USA	[23]
M 56 Fatal	Fentanyl p-FF Acetyl Fentanyl 4-ANPP Norfentanyl	B 62, U+ B 5.5, U+ U + U + U +	Xylazine U +, BE B 586, U +, Morphine U +, Cannabinoids U + Tramadol U +, Levamisole U +, Cocaine U +, CE U +	2021, USA	[23]
M 63 Fatal	Fentanyl p-FF Acetyl Fentanyl 4-ANPP Eutylone	B 8.8, U + U + U + U + B 2400, U +	Acetaminophen B 29,000, BE B 2029, U+, Ibuprofen U +, Acetaminophen U +, Levamisole U +, THC-COOH U +	2021, USA	[23]
M 44 Fatal	Fentanyl p-FF Acetyl Fentanyl 4-ANPP Norfentanyl	B 19, U + U + 4.9 U + U +	Tramadol B < 500, U+, Cocaine B < 50, U +, BE B 906, U +, Levamisole U +, CE U +	2021, USA	[23]
F 30 Fatal	Fentanyl p-FF Acetyl Fentanyl 4-ANPP Norfentanyl	B 6.8, U + U + U + U + U +	Xylazine U +, Meth B < 100, U +, Tramadol Volatiles B 0.48, U 0.56, HV 0.52	2021, USA	[23]
F 40 Fatal	Fentanyl p-FF Acetyl Fentanyl 4-ANPP Norfentanyl	B 30, U + U + U + U + U +	Xylazine U +, Diazepam B < 50, Nordiazepam B 78, U +, Meth B 260, U +, Amphetamine B < 100, U +, Fluoxetine U +, Temazepam U +, Oxazepam U +	2021, USA	[23]
M 33 Fatal	Fentanyl p-FF 4-ANPP Norfentanyl	B 22, U + U + U + U +	Meth B 2 000, U +, Amphetamine B 110, U +, Tramadol U +, Citalopram U +	2021, USA	[23]
6 Fatal	3-CMC			2021, EU	[31]
2 Intox 1 Fatal	3-MMC 3-MMC			2021, EU	[32]
NA 20–50 Fatal	Etonitazepyne	B +		2021, USA	[47]
M 42 Fatal	Butonitazene Metonitazene 4′-OH-nitazene 5-Amino metonitazene *N*-Desethyl metonitazene NEP	BP 3.2, S 2.4, U 10 BP 33, S 18, U 8.4 BP ND, S detected, U 9.8 BP 155%, S 14%, U 41% BP < 5%, S < 5%, U < 5% B +		2021, USA	[48]
M 19 Fatal	Metonitazene 4′-OH-nitazene 5-Amino metonitazene *N*-Desethyl metonitazene Etizolam α- Hydroxyetizolam	BP (1)8.7, BP (2) 7.6 BP (1) +, BP (2) + BP 15% BP 6% B 6.3 B 2.3	Tramadol B 1100, O-Desmethyltramadol B 270, OH-THC B 32, THC-COOH B 200, THC B 48, Caffeine B +, Naloxone B +	2021, USA	[48]
M 32 Fatal	Metonitazene 4′-OH-nitazene 5-Amino metonitazene *N*-Desethyl metonitazene *N*,*N*-Desethyl metonitazene Fentanyl Norfentanyl 4-ANPP	BP 10, U 28, VH est 3.7 U 10 BP 239% BP 20%, U 240%, VH 75% U 7% B 3.2 B 1.1 B +	Meth B 3900, Amphetamine B 160, Caffeine B +, Cotinine B +, Quinine B +	2021, USA	[48]
M 26 Fatal	Metonitazene 4-OH-nitazene Fentanyl Norfentanyl p-FF 4-ANPP	BC 1.6, U + U + B 12 B 0.66 B + B +	6-AM B 2.7, Morphine B 43, THC-COOH 20, Diphenhydramine B 460, Caffeine, Quinine, Ethanol 3.47 mmol/L	2021, USA	[48]
M 34 Fatal	Metonitazene 7-amino clonazepam	BP 0.52 B +	Meth B 1400, Amphetamine B 96, Alprazolam B 5, Diphenhydramine B 53, Citalopram/Escitalopram 420, Ethanol 3.25 mmol/L	2021, USA	[48]
M 58 Fatal	Flunitazene Metonitazene 4′-OH-nitazene 5-Amino metonitazene *N*-Desethyl metonitazene *N*,*N*-Didesethyl metonitazene 8-Aminoclonazolam Flualprazolam Fentanyl Norfentanyl 4-ANPP	BP 0.58, U 9.1 BP > 0.1, U 16 BP ND, U 18 BP > 300%, U 10% U 147% U 15% B + B + B 13 B 1.5 B +	Caffeine B +, Diphenhydramine B 300, Quinine B +	2021, USA	[48]
M 52 Fatal	Metonitazene 4′-OH-nitazene 5-Amino metonitazene	B 3.1 BP ND BP 23%	Caffeine B +, Ethanol 43.19 mmol/L	2021, USA	[48]
M 63 Fatal	Flunitazene Metonitazene 4′-OH-nitazene *N*-Desethyl metonitazene 8-Aminoclonazolam 4-ANPP Fentanyl Norfentanyl	B + BC < 0.5, U 0.58, VH est 1.1 BC ND, U+ VH + BC ND, U 117%, VH 14% B + B + B 7.5 B 0.85	Naloxone B, Gabapentin B 6.8 µg/mL, Caffeine B +, Diphenhydramine B200, Quinine B +	2021, USA	[48]
M 42 Fatal	Metonitazene 4′-OH-nitazene 5-Amino metonitazene *N*-Desethyl metonitazene Fentanyl Norfentanyl 4-ANPP	BP 8.9, U 14, VH + BP +, U 8.0, VH + BP 117% BP < 5%, U 72% B 17 B 3.8 B +	Caffeine B +, Quinine B +	2021, USA	[48]
M 40 Fatal	Metonitazene 4′-OH-nitazene 5-Amino metonitazene *N*-Desethyl metonitazene Fentanyl Norfentanyl Acetylfentanyl 4-ANPP	BP 2.3, U 4.6, HV + BP +, U 1.2 BP 269% U 72% B 5.8 B 1.2 B 0.49 B +	Meth B 29, Caffeine B +, Cotinine B +, Venlafaxine B 1300, *O*-desmethylvenlafaxine 390, Quinine B +	2021, USA	[48]
M 59 Fatal	Metonitazene 4′-OH-nitazene 5-Amino metonitazene *N*-Desethyl metonitazene Fentanyl Norfentanyl 4- ANPP	BP 2.4, U 46, VH est 1.8 BP 1.4, U 5.3 BP 197% BP 30%, U 36%, VH 295% B 33 B 10 B +	Morphine B 41, Caffeine B +, Cotinine B +, Gabapentin B 31 µg/mL, Fluoxetine B 85, Norfluoxetine B 46, Quinine B +	2021, USA	[48]
M 44 Fatal	Metonitazene 4′-OH-nitazene *N*-Desethyl metonitazene Fentanyl Norfentanyl 4-ANPP	BP 1.5, U 4.7, VH + U 2.7 U 17% B 16 B 1.2 B +	Meth B 16, Amphetamine B 6.6, Caffeine B +, Cotinine B +, Xylazine B +, Quinine B +, Ethanol 18.45 mmol/L	2021, USA	[48]
M 27 Fatal	Metonitazene 4′-OH-nitazene 5-Amino metonitazene *N*-Desethyl metonitazene 8-Aminoclonazolam Pyrazolam	BP 3.5, U 19, VH est 1.6 BP +, U 4.6, VH + BP 73%, U 6% U 70% B + B 14	Quinine B +, Caffeine B +, Ethanol 2.82 mmol/L	2021, USA	[48]
F 29 Fatal	Metonitazene 4′-OH-nitazene 5-Amino metonitazene *N*-Desethyl metonitazene	BP 13, U 10 VH est 1.0 BP +, U 2.1, VH + BP 51%, U 58%	Meth B 150, Amphetamine B 32, Caffeine B +, Cotinine B +, Naloxone B +, Nicotine B +, Mirtazapine B 160	2021, USA	[48]
M 35 Fatal	Metonitazene 4′-OH-nitazene 5-Amino metonitazene *N*-Desethyl metonitazene *N*,*N*-Didesethyl metonitazene	BP 5.8, U 4.0, VH est. 0.71 U 28, VH + BP 152%, U 62% BP 7%, U 294% U 46%	Caffeine B +, Cotinine B+, Mirtazapine B 37	2021, USA	[48]
M 47 Fatal	Flunitazene Metonitazene 4′-OH-nitazene 5-Amino metonitazene *N*-Desethyl metonitazene 8-Aminoclonazolam Flualprazolam Fentanyl Norfentanyl 4-ANPP	BP 2.1, CB 4.8, U 0.5 BP 5.0, CB 12, U 2.1 BP +, CB ND, U 0.5 BP 352% BP < 5%, U 20% B + B + B 3 B 0.44 B +	THC B 0.52, THC-COOH 12, Caffeine B +, Cotinine B +, Nicotine B +, Bupropion B 300, Hydroxybupropion 290, 10- Hydroxycarbazepine 9.5 µg/mL, Quetiapine B 590, Gabapentin 34 µg/mL	2021, USA	[48]
M 47 Fatal	Metonitazene 4′-OH-nitazene 5-Amino metonitazene N-Desethyl metonitazene Fentanyl Norfentanyl Acetylfentanyl 4-ANPP	BP 4.0, VH est. 0.76 VH + PB 44%, BP 15%, VH 97% B 41 B 2.4 B 25 B +	Caffeine B +, Naloxone B +, Diphenhydramine B 72, Quinine B +, Ethanol 4.55 mmol/L	2021, USA	[48]
M 32 Fatal	Flunitazene Metonitazene Fentanyl 4-ANPP	BC 0.6, U + BC 2.5, U 2.0 B 6.6 B +	Caffeine B +, Lorazepam B 24, Trazodone B 0.083 µg/mL, Ziprasidone B 10, Diphenhydramine B 110, Quinine B +, Ethanol 36.9 mmol/L	2021, USA	[48]
M 33 Fatal	*N*-ethylpentylone	B 64.7, U 2590, VH 279, Bile 2250, Brain 362 ng/g, K 289 ng/g, L 152 ng/g, G 2980	Amphetamine B 4.1, U 235, VH 9.2, B 106, Brain 33.8 ng/g, K 25.5 ng/g, L 25.3 ng/g, G 77.4	2021, Poland	[64]
M 38 Fatal	*N*-ethylpentylone	B 95.7, U 8580, VH 261, Bile 636, Brain 217, K 254 ng/g, L 260 ng/g, G 220	Amphetamine B 3, U 438, VH 9.2, B 22.9, Brain 15.4 ng/g, K 21.9 ng/g, L 14.5 ng/g, G 6.7	2021, Poland	[64]
M 38 Fatal	*N*-ethylpentylone	B 20.1, VH 133, Bile176, Brain 128, K 37.4 ng/g, L 68.7 ng/g	Amphetamine B 1.9, VH 4.5, B 40.1, Brain 18.4 ng/g, K 8.1 ng/g, L 20.2 ng/g	2021, Poland	[64]
F 22 Fatal	α-PiHP 4-CMC	B 6.1, U 31.7, VH 2.5, Brain 7.8 ng/g, G 246 B 4.2, U 437, VH 13.9, Brain 2.1 ng/g, L 2.1 ng/g, G 57.3	*O*-Desmethyltramadol B 1.2, U 45, VH 2.4, Brain 3.3 ng/g, L 3 ng/g, G 6.1, cis-Tramadol B 1.2, U 45, VH 2.4, Brain 3.3 ng/g, L 3 ng/g, G 6.1, *N*-Desmethyltramadol B 0.5, U 35, VH 1.7, Brain 1.1 ng/g, L 1.4 ng/g, G 3.5	2021, Poland	[64]
M 35 Fatal	4-MEC	BC 43.4 µg/mL, VH 2.9–4 µg/mL, U 619 µg/mL, B 43.5 µg/mL, G 28.2 µg/mL, U 619 µg/mL		2021, France	[49]
M 31 Intox	3-MMC	B 177, U 20,000	GHB B 131 mg/mL, U 2000 mg/mL, Ethanol B 12.58 mmol/L, U 16.49 mmol/L	2021, France	[50]
M 44 Fatal	3-MeO-PCP	B 525, U 384	Matrix not indicated Methadone 94, EDDP 16, THC 0.6, THC-COOH 8.6	2021, France	[65]
M 28 Intox	DCK Eutylone Flubromazolam Mitragynine 7-hydroxymitragynine	+ + + + +	Codeine U +, 7-aminoclonazepam U +, Hydrocodone U +, Hydromorphone U +, Lorazepam U +, Morphine U +, Oxycodone U + Oxymorphone U +, PCP U +, Desalkylflurazepam U +, Psilocin U +, O-desmethyltramadol U +, Cannabinoids U +	2021, USA	[51]
M 37 Intox	2 F-DCK CMC 3-MeO-PCP 3-OH-PCP N-ethylhexedrone	U 147 mg/L U 48 mg/L U 1100 mg/L U 12,085 mg/L U 165 mg/L	Clomipramine U +, Citalopram U +, Mianserine U +, Benzodiazepines U + Buprenorphine U +	2021, France	[52]
M NA Fatal	Isotonitazene	BP 2.28, BC 1.70, U 1.88, HV 0.36, Pericardiac Fluid 6.70, Lungs 0.52 ng/mg, L 0.04 ng/mg, K 1.61 ng/mg, Heart 7.74 ng/mg, Brain 18.6 ng/mg, Spleen 4.40 ng/mg, Mu 1.15 ng/mg, H 75 ng/mg	Diazepam B 29; Nordiazepam B 71; Oxazepam B 4.8; Mefenamic acid B 5.0 µg/mL, Domperidone B 6.0; Acetaminophen B 4.8 µg/mL	2021, Swiss	[53]
M NA Fatal	Isotonitazene	BP 0.59, BC 1.13, U 3.37, HV 0.12, Pericardiac Fluid 5.01, Lungs 17.9 ng/mg, L 0.04 ng/mg, K 1.02 ng/mg, Heart 2.17 ng/mg, Brain 2.72 ng/mg, Spleen 3.44 ng/mg, Mu 2.08 ng/mg, CSF 0.88, H 182 ng/mg	Lorazepam B 12; THC B 56; THC-OH B 1.8 THC-COOH B 6.5; CBN B 2.9	2021, Swiss	[53]
M NA Fatal	Isotonitazene	BP 0.74, BC 0.70, U 0.19, HV 0.65, Pericardiac Fluid 2.66, Lungs 2.39 ng/mg, L 0.04 ng/mg, K 0.67 ng/mg, Brain 4.45 ng/mg, Spleen 2.62 ng/mg, Mu 1 ng/mg, H 32–35 ng/mg (0–3 cm/3–6 cm)	ethanol B 12.37 mmol/L	2021, Swiss	[53]
M 18 Fatal	Etonitazepyne Flualprazolam Flubromazepam Hydroxy-flubromazepam	BP +, U + PB +, U + PB +, U + PB +, U +	Methadone PB +, EDDPP B +, Diazepam PB +	2021, UK	[39]
F 61 Fatal	Brorphine Fentanyl 4-ANPP	CB 2.0 CB 0.32 CB +	Ethanol CB +, Gabapentin CB +, Chlorpromazine CB +, Benzodiazepines CB +, Amphetamines CB +	2021, USA	[40]
M 30 Intox	Fentanyl Norfentanyl 4-ANPP	BP 303± 33 BP + BP +	Meth BP + Amphetamines BP, Xylazine BP 119 ± 11	2021, USA	[41]
M 15 Intox	5F-ADB O-desmethyl-5F-ADB 5-OH-pentyl-(*O*-desmethyl-)-ADB	B 0.11–3.9 µg/L (2 h) B + (2 h), U + (18, 38 h) B + (2 h), U + (18 h)		2021, France	[42]
M 23 Fatal	2-MAPB	BP 7.3 µg/mL, U 167 µg/mL, G 98.9, BC 16.7 µg/mL	THC BP 0.044, THC-COOH BP 0.067 µg/mL, U 0.138 µg/mL, G 0.287, BC 0.153 µg/mL, Diazepam BP 0.02 µg/mL, U 0.013 µg/mL, G 0.012, BC 0.018 µg/mL, Nordiazepam BP 0.01 µg/mL, U 0.02 µg/mL, G 0.012, BC 0.024 µg/mL, Temazepam BP 0.005 µg/mL, U 0.126 µg/mL, G 0.224, BC 0.005 µg/mL, Flephedrone BP 0.008 µg/mL, U 0.009 µg/mL, CB 0.003 µg/mL, 2C-B U 0.74 µg/mL	2021, German	[43]
M 22 Fatal	4F-MDMB-PINACA	U +, S +	THC S 1.4, THC-OH S 0.3, THC-COOH S 2.8	2021, Belgium	[44]
M 36 Fatal	Fentanyl Norfentanyl 4-ANPP	BC 44, BP 30, U 20, B 112, G 101, H 652 pg/mg BP 2, U 2, B10, G 1, H 92 pg/mg BP 0.6, U 0.2, B 1.6, G 0.3, H 9.5 pg/mg		2021, France	[45]
M 42 Fatal	2F-DCK 5-Meo-PCE 5-MeO-DMT	BP 1780, U 6106, Bile 12,200, VH 1500, H 4410–5080 pg/mg BP 90, U 6106, Bile 3500, VH 66, H 1610–3610 pg/mg BP 52, U 2190, Bile 1740, VH 155, H 1990–3390 pg/mg	THC PB 9.4, THC-COOH PB 24, U +, OH-THC PB 1.1, Amphetamine BP 27, U 970, Bile 43, Cocaine U +, HV +, BE BP 2.5, U 787, Bile 32,800, HV +, EME BP 9.8, U 32, Bile 8400, HV +, Levamisole PB +, U +, VH +. Lorazepam U 7.5, Bile 186	2021, France	[68]
M 27 Fatal	Furanyl fentanyl	B 11.2		2020, Poland	[22]
F 30 Fatal	Fentanyl p-FF Acetyl fentanyl Norfentanyl 4-ANPP	B 26, U + U + U + U + U +	Amphetamines B > 100, U Amphetamine U +, Tramadol U +	2020, USA	[23]
F 54 Fatal	Fentanyl p-FF Norfentanyl 4-ANPP	B 4.1 U + U + U +	Cocaine B 60, U +, CE B 67, U +, BE B 677, U + Volatiles B 0.14, U 0.17, HV 0.16 g/dL	2020, USA	[23]
M 29 Fatal	Fentanyl p-FF Acetyl fentanyl 4-ANPP	B 11, U + > 2.5, U + U + U +	Xylazine U +, Lidocaine B +, U +, Meth B 730, U +, Amphetamine B 280, U +, Morphine U +	2020, USA	[23]
F 35 Fatal	Fentanyl p-FF Acetyl fentanyl 4-ANPP Norfentanyl	B 14, U + U + U + U + U +	Alprazolam B 14, BE B 168, U +, Nordiazepam B > 50, U +, Meth B 640, U+ Amphetamine B > 100, U +, α-OH-alprazolam U +, Cocaine U +, Oxazepam U +, Temazepam U +, Levamisole U +	2020, USA	[23]
3 Fatal	3-CMC			2020, Europe	[31]
5 Fatal	3-MMC			2020, Europe	[32]
M 32 Fatal	Eutylone	BC (postmortem) 4.29 ng/g, U 192 ng/g, G 2.120 ng/g, Fat Tissue 1.31 ng/g, BP 2500	Aripiprazole BC 49.1 ng/g, U 34.5 ng/g, Fat Tissue 358 ng/g, BP 26.7 ng/g	2020, Japan	[33]
M 57 Fatal	MDMB-4en-PINACA 3,3-dimethylbutanoic acid	BP +	Ethanol 18.2 mmol/L	2020, USA	[34]
NA DUID Intox	MDMB-4en-PINACA MDMB-4en-PINACA 3,3-Dimethylbutanoic acid 4F-MDMB-BINACA 5F-MDMB-PICA 5-OH-MDMB-PICA 5F-MDMB-PICA 3,3-dimethylbutanoic acid	B + B + B + B + B + B +		2020, USA	[34]
NA Fatal	MDMB-4en-PINACA 4F-MDMB-BINACA	BP + BP +		2020, USA	[34]
M 27 Fatal	MDMB-4en-PINACA 4F-MDMB-BINACA 4-OH-MDMBBINACA 5F-MDMB-PICA 5-OH-MDMB-PICA	B + B + B + B + B +	Cotinine B +	2020, USA	[34]
NA Fatal	MDMB-4en-PINACA 4F-MDMB-BINACA 4-OH-MDMBBINACA	BP + BP + BP +		2020, USA	[34]
NA Fatal	MDMB-4en-PINACA MMB-FUBINACA 3-methylbutanoic acid	BP + BP +		2020, USA	[34]
NA Fatal	MDMB-4en-PINACA	CB +		2020, USA	[34]
M 57 Fatal	4-MDP Dihydro-4-MEC *N*-Deethyl-4-MEC *N*-Deethyl-dihydro-4-MEC	BP 1285, CB 1128, B 1187, VH 734–875, U > 10,000 B + B + B +	Cocaine B +, Sildenafil B +, Bromazepam B +, Nevirapine B +, Benzodiazepines B +	2020, France	[35]
M 21 Fatal	*N*-ethylexedrone	B 145	Amphetamine B 12, THC-COOH B < 5	2020, Poland	[36]
11 Intox	MDMB-4en-PINACA			2020, UK	[37]
5 Intox 21 Fatal	4F-MDMB-BICA 4F-MDMB-BICA			2020, UK 2020, Hungary	[61]
F 33 Fatal	MDMB-4en-PINACA 4F-MDMB-BINACA Fentanyl	B + B + B 11	Xylazine B 6.1 ng/mL, Diazepam B 29, Morphine B 53, Naloxone B +, Cotinine B +	2020, USA	[34]
M 42 Fatal	Eutylone 5F-MDMB-PICA 3,3-Dymethylbutanoic Acid	BP 1.2 U +	Ethanol U 47.75 mmol/L	2020, USA	[34]
M 37 Fatal	Eutylone Fentanyl 4-ANPP	BP 110 BP 26 BP 4.6	BE BP 711	2020, USA	[24]
M 17 Fatal	Eutylone	BP 330	Ethanol 31.9 mmol/L, THC 1.1, THC-COOH 24	2020, USA	[24]
M 20 Fatal	Eutylone	CB 1.6	Alprazolam BC 66, Cocaine BC 228, BE BC 411, Ketamine BC +, Etizolam BC 89	2020, USA	[24]
M 1D Fatal	Eutylone	BP 1.9	B Venlafaxine < 500	2020, USA	[24]
M 42 Fatal	Eutylone	BP 3.1		2020, USA	[24]
M 39 Fatal	Eutylone	BP 4.4	THC-COOH U +, Cocaine U +, BE U +	2020, USA	[24]
M 53 Fatal	Brorphine Flualprazolam Fentanyl Norfentanyl 4-ANPP Isotonitazene	B 10, U 23 Matrix not indicated 50 Matrix not indicated 3.4 Matrix not indicated 0.36 Matrix not indicated + Matrix not indicated +	Matrix not indicated Caffeine +, Cotinine +, Quinine +, Codeine 6.6, Morphine, 6-AM 1.5, Citalopram/Escitalopram 76	2020, USA	[25]
M 60 Fatal	Brorphine Fentanyl Norfentanyl 4-ANPP Flualprazolam	B 0.9, U 0.4 Matrix not indicated 14 Matrix not indicated 11 Matrix not indicated + Matrix not indicated +	Matrix not indicated Caffeine +, Methadone 160, EDDP 45, Morphine 42, Bupropion, Hydroxybupropion 380, Duloxetine 520, Lamotrigine 6.8 µg/mL, Gabapentin 15 µg/mL	2020, USA	[25]
M 45 Fatal	Brorphine Flualprazolam 4-ANPP Fentanyl	B 1, U 1.9 Matrix not indicated 2.5 Matrix not indicated + Matrix not indicated 5	Matrix not indicated Caffeine +, Cotinine +, Tramadol 33 ng/mL, THC 0.62,	2020, USA	[25]
F 57 Fatal	Brorphine Fentanyl Norfentanyl Acetyl Fentanyl 4-ANPP	B + Matrix not indicated 31 Matrix not indicated 5.5 Matrix not indicated 0.13 Matrix not indicated +	Matrix not indicated Caffeine +, Cotinine +, Naloxone +, Cocaine 110, BE 1300, THC 0.88, Diphenhydramine 61, Meth 730,	2020, USA	[25]
M 42 Fatal	Brorphine Fentanyl Norfentanyl 4-ANPP Flualprazolam	B 1.1, U 3.3 Matrix not indicated 36 Matrix not indicated 1.4 Matrix not indicated + Matrix not indicated +	Matrix not indicated Caffeine +, Cotinine +, Naloxone +, Diphenhydramine 620, Morphine 110, 6-AM 7.3	2020, USA	[25]
M 60 Fatal	Brorphine Fentanyl Flualprazolam	B 8.1, U 21 Matrix not indicated 3.1 Matrix not indicated +	Matrix not indicated Cotinine +, Sertraline 26, Desmethylsertraline 110, Verapamil 42, Diphenhydramine 960, Morphine 79, 6-AM 2.5	2020, USA	[25]
M 47 Fatal	Brorphine Fentanyl Norfentanyl 4-ANPP	B 2.5 Matrix not indicated 16 Matrix not indicated 1.1 Matrix not indicated +	Matrix not indicated Caffeine +, Cotinine +, Naloxone +, Cocaine 110, BE 1300, THC Naloxone, Oxycodone 22, Sildenafil 35, N-Desmethylsildenafil 10	2020, USA	[25]
M 39 Fatal	Brorphine Fentanyl Norfentanyl 4-ANPP Flualprazolam	B 6.7, U 7.3 Matrix not indicated 45 Matrix not indicated 2.1 Matrix not indicated + Matrix not indicated +	Matrix not indicated Caffeine +, Cotinine +, Nicotine +, Alprazolam 14, Tramadol 70, Gabapentin 10 µg/mL, Diphenhydramine 1200, Codeine 6.5, Morphine 290, Hydromorphone 4.7	2020, USA	[25]
F 37 Fatal	Brorphine Fentanyl 4-ANPP	B 0.7 Matrix not indicated 22 Matrix not indicated +	Matrix not indicated Ethanol 30 mmol/L, Cotinine +, Naloxone +, THC-COOH 85, THC 18, Diphenhydramine 110	2020, USA	[25]
M 48 Fatal	Brorphine Flualprazolam Fentanyl Norfentanyl Acetyl Fentanyl 4-ANPP	B 0.6, U 0.2 Matrix not indicated 5.4 Matrix not indicated 4.7 Matrix not indicated 1.6 Matrix not indicated 1.2 Matrix not indicated +	Matrix not indicated Caffeine +, Naloxone +, Morphine 8, Diphenhydramine 190, Clonazolam +	2020, USA	[25]
F 47 Fatal	Brorphine Flualprazolam 4-ANPP Fentanyl Norfentanyl Acetyl Fentanyl	B 6.7, U 2.1 Matrix not indicated 13 Matrix not indicated + Matrix not indicated 190 Matrix not indicated 5.4 Matrix not indicated 0.15	Matrix not indicated Cotinine +, Naloxone +, Codeine 7, Morphine 85, 6-AM 12, Xylazine 170, Amphetamine 55, Meth 580	2020, USA	[25]
F 30 Fatal	Brorphine Fentanyl Norfentanyl	B 0.3, U 1.4 Matrix not indicated 0.37 Matrix not indicated 0.97	Matrix not indicated Caffeine +, Naloxone +, Midazolam 20, Amphetamine 110, Meth 1900, MDA 9.8, MDMA 75,	2020, USA	[25]
M 53 Fatal	Brorphine Flualprazolam 4-ANPP Fentanyl Norfentanyl	U 0.2 Matrix not indicated 20 Matrix not indicated + Matrix not indicated 19 Matrix not indicated 4.2	Matrix not indicated Caffeine +, Naloxone +, Morphine 15, Xylazine 30	2020, USA	[25]
M 57 Fatal	Brorphine Norfentanyl Acetyl Fentanyl Flualprazolam Isotonitazene 4-ANPP Fentanyl	B 0.5 Matrix not indicated 20 Matrix not indicated 0.1 Matrix not indicated + Matrix not indicated + Matrix not indicated + Matrix not indicated 130	Matrix not indicated Cotinine +, Naloxone +, Tramadol 48, Diphenhydramine 950, Morphine 21	2020, USA	[25]
F 54 Fatal	Brorphine Fentanyl Flualprazolam	B 0.1 Matrix not indicated 17 Matrix not indicated +	Matrix not indicated Ethanol 4.12 mmol/L, Codeine 21, Morphine 290, 6-AM 34, Lamotrigine 0.60 µg/mL, Topiramate 9400, Cyclobenzaprine 38, Amphetamine 140	2020, USA	[25]
M NA Fatal	Brorphine Fentanyl 4-ANPP	B 0.7 Matrix not indicated 32 Matrix not indicated +	Matrix not indicated Ethanol 100, Caffeine +, Cotinine +, Naloxone +, Nicotine +, Nordiazepam 130, Chlordiazepoxide 66, Lorazepam 9.4, THC 0.70, Diphenhydramine 53	2020, USA	[25]
M 51 Fatal	Brorphine Fentanyl Norfentanyl Flualprazolam 4-ANPP	B 1.1, U 0.4 Matrix not indicated 9.3 Matrix not indicated 5.6 Matrix not indicated + Matrix not indicated +	Matrix not indicated Ethanol 21.7 mmol/L, Caffeine +, Cotinine +, Naloxone +, Nicotine +, Cocaine 71, Benzoylecgonine 1600, Diphenhydramine 98, Morphine 19	2020, USA	[25]
F 49 Fatal	Brorphine Norfentanyl Acetyl Fentanyl Fentanyl Flualprazolam 4-ANPP	B 3.8, U 0.8 Matrix not indicated 12 Matrix not indicated 2.0 Matrix not indicated 21 Matrix not indicated + Matrix not indicated +	Matrix not indicated Caffeine +, Naloxone +, Diphenhydramine 260, Morphine 70	2020, USA	[25]
M 29 Fatal	Brorphine Flualprazolam 4-ANPP Fentanyl Norfentanyl	B 1.1, U 0.8 Matrix not indicated 3.6 Matrix not indicated + Matrix not indicated 37 Matrix not indicated 1.3	Matrix not indicated Caffeine +, Cotinine +, Naloxone +, Nicotine +, Quinine +, Acetaminophen 16 mcg/mL, 7-Amino Clonazepam 5.2 ng/mL, Tramadol 70, Diphenhydramine 490, Amphetamine 10, Meth 42	2020, USA	[25]
M 61 Fatal	Brorphine Fentanyl Norfentanyl 4-ANPP	B 0.4, U 0.2 Matrix not indicated 21 Matrix not indicated 1.9 Matrix not indicated +	Matrix not indicated Ethanol 12.37 mmol/L, Cotinine +, Naloxone +, Nicotine +, Alprazolam 65 ng/mL, Benzoylecgonine 330, Morphine 33, 6-AM 2.3, Gabapentin 9.9 µg/mL	2020, USA	[25]
F 27 Fatal	Isotonitazene Fentanyl Norfentanyl 4-ANPP Etizolam	B 1 B 5.7 B2.4 B 1.4 B 6.2	Diazepam B 120, Nordiazepam B 210, Oxazepam B 22, THC-COOH 64, THC 1.3, Caffeine B +, Cotinine B +, Naloxone B +, Diphenhydramine B +, Quinine B +	2020, USA	[26]
M 27 Fatal	Isotonitazene Etizolam	B 1.9, HV 0.1, U 2.6 B 15	Caffeine, B +, Acetaminophen B, Diphenhydramine B +	2020, USA	[26]
M 66 Fatal	Isotonitazene Fentanyl Norfentanyl Etizolam	B 1.5, U 0.6 B 2.9 B 1 B 13	Levetiracetam B 14 µg/mL, Diphenhydramine B 200, Cotinine B +, Quinine B +, Morphine U +	2020, USA	[26]
M 41 Fatal	Isotonitazene Fentanyl Norfentanyl Acetyl Fentanyl 4-ANPP	B 0.9, U 3.5 B 5.8 B 0.61 B 0.69 B 1.6	Morphine-free B 12, Cocaine B 89, Benzoylecgonine B 800, Naloxone B +, Tramadol B +, O-desmethyltramadol B +, Acetaminophen B +, Cotinine B +, Caffeine B +, Levamisole B +, Quinine B +, Xylazine U +	2020, USA	[26]
M 53 Fatal	Isotonitazene U-47700 4-ANPP	B 2.7 U 0.34 B +	BE 370, Phenacetin B +, Levamisole B +, Diphenhydramine B +	2020, USA	[26]
M 44 Fatal	Isotonitazene Etizolam	B 4.4, U 0.6 B 10	Hydromorphone-free B 3.3, Tramadol 670, O-desmethyltramadol B 310, Amphetamine B 65, Meth B 330, Diazepam B 150, Nordiazepam B 330, Oxazepam 22, Temazepam 20, 7-amino clonazepam B 29, Doxepin 290, Cotinine, Flualprazolam B +, U +	2020, USA	[26]
M 27 Fatal	Isotonitazene Etizolam	B 1.8, U 2.8 B 30	THC-COOH B 7.7, THC B 1.2, Diphenhydramine B 190, Caffeine B +, Cotinine B +, Piperidyl thiambutene B +, U +, Diphenhydramine B +, Quinine B +	2020, USA	[26]
M 56 Fatal	Isotonitazene Etizolam Flualprazolam	B 0.4 B + B 4.0	Naloxone, Hydroxyzine B +, cotinine B +, Caffeine B +, O-desmethylvenlafaxine B +, Venlafaxine B +	2020, USA	[26]
M 41 Fatal	Isotonitazene	B 1.7	Flualprazolam B 6.4, Caffeine B +, Cotinine B +	2020, USA	[26]
M 36 Fatal	Isotonitazene Flualprazolam	B 0.4 B 9.2	Naloxone B +, Alprazolam B 10, Amphetamine B 6.2, Meth B 5.9, MDA B 6.8, MDMA B 74, Caffeine B +, Cotinine B +, Hydroxybupropion, B +, Aripiprazole B +	2020, USA	[26]
M 48 Fatal	Isotonitazene Flualprazolam	B 1.8 B 5.3	Caffeine B +, Cotinine B +	2020, USA	[26]
M 24 Fatal	Isotonitazene Flualprazolam	B 2.2 B 10	THC B 1.7, THC-COOH B 6.8, Caffeine B +, Cotinine B +, Diphenhydramine B +	2020, USA	[26]
M 40 Fatal	Isotonitazene Etizolam	B 2.3 B 10	Hydrocodone B 5.3, Sertraline B 95, Desmethylsertraline B 170, Diphenhydramine B 97, Hydroxyzine B 64, Flualprazolam B +, Codeine B +, Methadone B +, Quinine B +, Quetiapine B +	2020, USA	[26]
M 42 Fatal	Isotonitazene Fentanyl Norfentanyl Acetyl Fentanyl 4-ANPP Flualprazolam	B 1.3 B 9 B 12 B 0.11 B + B +	Morphine-free B 72, Naloxone B +, Cocaine B 290, BE B 2400, Diphenhydramine B 340, Cotinine B +, Phencyclidine B +, Tramadol B +, Zolpidem B +, Quinine B +, Quetiapine B +	2020, USA	[26]
M 28 Fatal	Isotonitazene Fentanyl Norfentanyl 4-ANPP Mitragynine mCPP Flualprazolam	B 3.1 B 100 B 3.9 B + B 150 B + B 6.5	Morphine-free B 62, 6-AM-Free B 3.0, Naloxone B +, Cocaine B 96, BE 1,4, Sertraline B 66, Desmethylsertraline B 350, Diphenhydramine B 86, Caffeine B +, Cotinine B +, Piperidylthiambutene B +, Benzylfuranylfentanyl B +, Trazodone B +, Quinine B +	2020, USA	[26]
M 35 Fatal	Isotonitazene Flualprazolam	B 1.3 B 5.2	BE B 200, OH-THC B 1.3, THC-COOH 6.9, THC B 1.8, Ethanol 16.49 mmol/L, Caffeine B +, Cotinine B +, CE B +	2020, USA	[26]
M 46 Fatal	Isotonitazene Fentanyl Norfentanyl 4-ANPP Etizolam Flualprazolam	B 9.5 B 3.6 B 1.7 B 0.53 B + B +	Tramadol B 22, Naloxone B +, Diphenhydramine B 280, Cotinine B+, O-desmethyltramadol B +, Quinine B +	2020, USA	[26]
M 32 Fatal	Flualprazolam	B 2.1	Mitragynine, Cyclobenzaprine, Hydroxyzine, Delta-9 THC, Gabapentin and BZE	2020, USA	[27]
M 22 Fatal	Flualprazolam	B 3.2	Ethanol 37.55 mmol/L, Desmethylloperamide B +	2020, USA	[27]
M 36 Fatal	Flualprazolam	B 3.6	Meth B +, Amphetamine B +	2020, USA	[27]
M 29 Fatal	Isotonitazene Flualprazolam	B + B 4.1	Meth B +, Amphetamine B +	2020, USA	[27]
M 49 Fatal	Flualprazolam	B 4.5	BE +	2020, USA	[27]
M 38 Fatal	Isotonitazene Flualprazolam Fentanyl	B+ B 6.2 B+	Meth B +, Amphetamine B +, Hydroxyzine B +	2020, USA	[27]
M 21 Fatal	Flualprazolam	B 29	Amphetamine B +	2020, USA	[27]
M 21 Fatal	Isotonitazene Flualprazolam	B+ B 96	Diazepam B +, THC B +	2020, USA	[27]
M 42 Fatal	Flualprazolam Fentanyl 4-ANPP	B 110 B + B +	Diazepam B +, Cocaethylene B +, BZE B +, THC B +, Ethanol 28.43 mmol/L	2020, USA	[27]
M 40 Fatal	α-PVP	BP 0.81 mg/L	Amphetamine PB 0.34 mg/L, Alprazolam PB +, Oxazepam PB +	2020, Finland	[62]
M 24 Intox	Brorphine	S 10–20		2020, Belgium	[28]
M 29 Fatal	5-MeO-DIPT 5-OH-DIPT 5-MeO-IPT	U 2.79 U + U +		2020, China	[29]
M 31 Fatal	5-MeO-DIPT 5-OH-DIPT 5-MeO-IPT	U < 2 U + U +		2020, China	[29]
M 59 Fatal	5-MeO-DIPT 5-OH-DIPT 5-MeO-IPT	U < 2 U + U +		2020, China	[29]
M 24 Fatal	5-MeO-DIPT 5-OH-DIPT 5-MeO-IPT	U < 2 U + U +		2020, China	[29]
M 43 Fatal	5-MeO-DIPT 5-OH-DIPT 5-MeO-IPT	U < 2 U + U +		2020, China	[29]
M 22 Fatal	5F-AKB-48 5F-AKB-48 desfluoro-hydroxypentyl metabolite AB-FUBINACA 5F-PB22 5F-PB22-desfuoro-hydroxypentyl metabolite 5F-PB22-Et	B 9.5 μg/L B 294 μg/L B 1.7 μg/L B 6.2 μg/L B 0.3 μg/L B 0.4 μg/L	Ethanol B 46.01 mmol/L, U 61.42 mM/L	2020, UK	[30]
M 60 Fatal	Isotonitazene Flualprazolam	B 1.7 B +	Naloxone B, Cocaine B 29, CE B 23, BE B 470, Ethanol B 11.5 mmol/L, Cotinine B +, Diphenhydramine B +, Quinine B +	2020, USA	[26]

Definitions: +, positive; 2-FDCK, 2-Fluorodeschloroketamine; 2-MAPB, Methylaminopropylbenzofuran; 3-CMC, 3-chloromethcathinone; 3-Meo-PCP, 3-Methoxyphencyclidine; 3-MMC, 3-methylmethcathinone; 3-OH-PCP, 3-hydroxyphencyclidin; 4-ANPP, 4-anilino-N-phenethylpiperidine; 4-CMC, 4-chloromethcathinone; 4-FIBP, 4-fluoroisobutyryl fentanyl; 4F-MDMB-BINACA, methyl 2-(1-(4-fluorobutyl)-1*H*-indazole-3-carboxamido)-3,3-dimethylbutanoate; 4F-MDMB-PICA, methyl 2-{[1-(4-fluorobutyl)-1*H*-indole-3-carbonyl]amino}-3,3-dime-thylbutanoate; 4-MDP, 4-methylpentedrone; 4-MEC, 4-methylethcathinone; 4-OH-nitazene, 4-hydroxynitazene; 5F-ADB, *N*-[[1-(5-fluoropentyl)-1*H*-indazol-3-yl]carbonyl]-3-methyl-d-valine methyl ester; 5-F-ADB, methyl 2-(1-(5-fluoropentyl)-1H-indazole-3-carboxamido)-3,3-dimethylbutanoate; 5F-MDMB-PICA, Methyl-2-[[1-(5-fluoropentyl)indole-3-carbonyl]amino]-3,3-dimethyl-butanoate); 5-MeO-DIPT, 5-Methoxy-*N*,*N*-Diisopropyltryptamine; 5-Meo-IPT, 5-*N*-isopropyltryptamine; 5-OH-DIPT, 5-hydroxy-*N*,*N*-diisopropyltryptamine; 6-AM, 6-acetylmorphine; ADB-FUBINACA, *N*-(1-amino-3,3-dimethyl-1-oxobutan-2-yl)-1-(4-fluorobenzyl)-1*H*-indazole-3-carboxamide; B, whole blood; BE, Benzoylecgonine; BP, blood pheripheral; Br, Brain; CB, central blood; CMC, chloromethcathinone; CSF, cerebrospinal fluid; DCK, deschloroketamine; Dihydro-4-MEC, Dihydro-4-methylpentedrone; EDDP, 2-ethylidene-1,5-dimethyl-3,3-diphenylpyrrolidine; Fluoro-methyl-PVP, Fluoro-methyl-pyrrolidinovalerophenone; G, Gastric content; GHB, Gamma Hydroxybutyrate; HV, Vitrous Humor; K, Kidney; L, Liver; MDMB-4en-PINACA, Methyl-3,3-dimethyl-2-(1-(pent-4-en-1-yl)-1*H*-indazole-3-carboxamido)butanoate; MDMB-4en-PINACA, Methyl-3,3-dimethyl-2-(1-(pent-4-en-1-yl)-1*H*-indazole-3-carboxamido)butanoate; MDPHP, 3,4-methylenedioxy-α-pyrrolidinohexanophenone; Meth, Methamphetamine, ND, not detected; *N*-Deethyl-4-MEC, *N*-Deethyl-4-methylpentedrone; *N*-Deethyl-dihydro-4-MEC, *N*-Deethyl-dihydro-4-methylpentedrone; NEH, *N*-Ethylhexedrone; p-FF, parafenilfentanyl; S, serum; THC, Δ^9^-tetrahydrocannabinol; THC-COOH, 11-nor-9-carboxy-THC; THC-OH, 11-hydroxy-THC; U, urine; α-PIHP, alpha-pyrrolidinoisohexaphenone; α-PVP, alpha-pyrrolidinovalerophenone.

## 4. Discussion

### 4.1. Synthetic Opioids Related Intoxications and Deaths

Synthetic opioids [71] are the third largest class of NPS worldwide, increasing from 14 different substances in 2009 to 87 in 2020, of which 22 were reported for the first time. Synthetic opioids have a high abuse liability due to their binding affinity to μ-opioid receptors. Furthermore, their high drug lipophilicity facilitates crossing through the blood–brain barrier.

During the last decade, the most-abused synthetic opioids were fentanyl and its analogues. Many fentanyl analogues were scheduled starting in 2019 [72], leading to the development of benzimidazoles, an emerging synthetic opioid class that entered the black market as a cheaper alternative to heroin.

As widely described [73], intoxications and deaths involving synthetic opioids are of particular concern in the United States of America (USA). From 2013 to 2019, the death rate from synthetic opioids increased in the USA by 1040%. Consistently, throughout the COVID-19 pandemic, synthetic opioids were the NPS class [23,24,25,28,34,39,40,41,45,47,48,51,60,69] most involved in intoxications and fatalities.

The total number of single and mixed synthetic opioid cases reported during COVID-19 was 114, with 3 intoxications and 111 fatalities. Cases occurred [23,24,25,28,34,39,40,41,45,47,48,51,60,69] in the USA (n = 107), Switzerland (n = 3), Poland (n = 1), France (n = 1), Belgium (n = 1) and the UK (n = 1). Notably, DBZPs were also involved in 45 cases; flualprazolam and etizolam [47,48,73,74,75,76] were detected in 64% (n = 29) and 15% (n = 7) of the cases, respectively.

### 4.2. Fentanyl and Analogues

Synthetic opioids [77,78] induce euphoria, sedation and respiratory depression, with potencies 100 to 10,000 times greater than morphine, producing primarily fatal cases.

Despite fentanyl-related substances’ placement in the FDA’s Schedule I in 2018, during the COVID-19 pandemic, 64 deaths involving fentanyl were reported. Fentanyl and its analogues [79] continue to have a critical worldwide impact on drug intoxications and fatalities. Additionally, 90% of seized synthetic opioids were fentanyl.

The most frequent fentalogues reported with fentanyl were p-FF and acetyl fentanyl (and their metabolites), representing 43% of total cases (n = 43).

### 4.3. Other Synthetic Opioids

Brorphine [25,28,40], structurally similar to fentanyl, is a substituted piperidine benzimidazolone. Brorphine was reported in 9 deaths during the COVID-19 pandemic, with in vitro receptor activation assays showing a similar potency and efficacy to fentanyl.

### 4.4. New Synthetic Opioids (Benzimidazoles and AP-Series Drugs)

Among all synthetic opioids previously reported [80,81], a new wave of designer opioids raised concern worldwide, entering the market in 2019. Benzimidazoles, structurally unrelated to fentanyl, are synthetic analogues of etonitazene, discovered for the first time in Switzerland by a pharmaceutical company [80]. Recently, benzimidazoles were shown to have a considerable degree of affinity and a higher potency than that of fentanyl for the µ-opioid receptor.

The new synthetic opioids, benzimidazoles and cinnamoylpiperazines [26,47,48,60] subclasses were detected, both alone and in combination with other NPS classes. Respectively, one case was reported each for butonitazene, flunitazene (n = 4), metonitazene (n = 18), isotonitazene (n = 26), etonitazepyne (n = 8), 2-methyl AP-237 (n = 5) and AP-238 (n = 2). Isotonitazene (n = 15) and metonitazene [26,48] (n = 16) metabolites were reported in several cases. The specific role that benzimidazoles played in these deaths remains unclear. The presence of active isotonitazene metabolites [80] was suggested to indicate higher toxicity. Specifically, the activity of N-desethyl- and nor-isotonitazene metabolites was higher than that of isotonitazene.

The rapid placement of new substances into controlled drug schedules is one important action to prevent such substances from spreading out of control. Unfortunately, scheduling still remains a lengthy and time-consuming process. Most isotonitazene cases in the USA were reported before isotonitazene was scheduled in June 2020.

After fentanyl-related substances [60] were placed under control, the cinnamoylpiperazine subclass emerged onto the market. 2-Methyl AP-237 and AP-238 were recently reported in several cases (n = 7). This class of compounds [71], due to its piperazine ring, has highly potent analgesic effects. To date, few compounds in this class have been reported, but minimal structural modification could generate even more potent compounds that could easily lead to overdoses and deaths.

Currently, the mechanisms by which NPS affect cardiac function have not yet been elucidated. NPS inhibit monoamine transporters [11,82], potentially giving rise to indirect cardiovascular effects. However, NPS [11] may also have direct effects on cardiomyocyte function, irrespective of circulating norepinephrine levels.

### 4.5. Synthetic Cannabinoids

According to the European Drug Report [7], synthetic cannabinoids are the NPS class with the highest number of substances monitored by the EU Early Warning System in 2021. They mimic the effects of the natural constituent delta9-tetrahydrocannabinol (THC), which is the principal compound producing cannabispsychoactive effects. First synthetized for research purposes, synthetic cannabinoids are sold on the black market. Synthetic cannabinoids [83] are agonists at the CB1 and CB2 cannabinoid receptors, but also ion channels and PPAR receptors, where they can produce severe adverse effects, including disinhibition, euphoria, neurological disorders, agitation, irritability, paranoia, psychiatric conditions and death. There were 19 intoxications and 30 deaths published during the COVID-19 pandemic involving synthetic cannabinoid intake.

### 4.6. F-MDMB-BICA

4F-MDMB-BICA was the most commonly abused synthetic cannabinoid in 2020. Consumption of 4F-MDMB-BICA [37] was associated with a high number of intoxications and deaths in 2020 in the UK and Hungary (n = 5 and n = 21, respectively). No further cases were described after the end of 2020.

### 4.7. MDMB-4en-PINACA

MDMB-4en-PINACA [34] is a CB1 receptor agonist, with a potency three times higher than its analogue JWH-018. In 2020, There were 12 intoxications and 7 fatalities reported [34,37], 6 of which were in combination with other synthetic cannabinoids. While it has been available in the market since 2017, MDMB-4en-PINACA seizures increased during 2020 [61] after two similar analogues, 4F-MDMB-BINACA and 5F-MBMB-PICA, were placed under international control. This example confirmed the importance of rapid detection and scheduling, but also demonstrated that, as soon as a substance is controlled, a new analogue appears on the market.

### 4.8. Synthetic Cathinones

Synthetic cathinones [7] are the second-largest NPS class, based on the EU Early warning system (EWS). As of this writing, 162 synthetic cathinones are monitored by the EU EWS. They have a similar structure to the natural compound cathinone, principally found in *Catha edulis*, and amphetamines. Synthetic cannabinoids are b-keto analogues of a corresponding phenethylamine. During the COVID-19 pandemic, 42 cases {Formatting Citation} were reported with synthetic cannabinoids, alone or in combination with other NPS classes.

### 4.9. 3-CMC

3-CMC has psychostimulant effects similar to those of methcathinone and 4-chloromethcathinone (4-CMC), which are Schedule I and II of the 1971 United Nation Single Convention [84] on Psychotropic Substances, respectively.

As reported in the European Drug Report [7], 3-CMC accounted for almost a quarter of the total drug seizures. This increase in 3-CMC seizures could have been related to the recent control of 4-CMC under the international drug system.

In Europe, between 2020 and 2021, a total of 9 fatalities (6 in Poland and 3 in Sweden) were reported in the risk assessment published by the EMCDDA. In some cases, 3-CMC [31] was the only substance identified, but in other cases, multiple stimulants and depressants were also observed.

### 4.10. 3-MMC

3-MMC is a synthetic cathinone related to methcathinone and 4-methylmethcathinone (mephedrone, 4-MMC). 3-MMC was reported for the first time in Sweden in September 2012 [32]. The distinction between the two positional isomers requires hyphenated analytical techniques. 3-MMC [32] was reported in 6 fatalities and 2 intoxications. No specific data were available regarding these cases.

### 4.11. Designer Benzodiazepines

Recently [85], there was a significant increase in the number of new DBZPs. The mechanism of action on the central nervous system is via the GABA_A_ receptor. GABA induces inhibitory effects through the central nervous system, including hypnosis, sedation or amnesia. DBZPs have a high abuse potential. Between 2020 and 2021 [14] clonazolam, diclazepam, etizolam, flualprazolam and flubromazolam were included in Schedule IV of the Convention of Psychotropic Substances of 1971.

Flualprazolam was detected alone in only 5 cases and, as mentioned in “Single NPS class Fatalities” [23,27,28,41,45,60], most of the cases reported during the pandemic period involved DBZPs (flualprazolam and etizolam) co-consumed with SOs.

### 4.12. Tryptamines & Arylcyclohexylamines & Phenethylamines

Tryptamines [86] are a psychedelic derived from the decarboxylation of the amino acid tryptophan, also known as indolealkylamines. Slight modifications to the indolealkylamine backbone yield a huge panel of novel tryptamine structures. Some easily cross the blood–brain barrier and have highly potent effects. There are few published data on the tryptamines, which could lead to underestimation of the level of tryptamine consumption and abuse. During the COVID-19 pandemic, only 5 fatalities [29] were reported, involving 5-Methoxy-*N*,*N*-Diisopropyltryptamine (5-(MeO-DIPT), 5-hydroxy-*N*,*N*-diisopropyltryptamine, (5-OH-DIPT) and 5-N-isopropyltryptamine (5-MeO-IPT).

Arylcyclohexylamines and phenethylamines have similar mechanisms of action to tryptamines and elicit stimulating effects; hence, they are included in the same NPS subgroup. Only 2 cases were reported for phenethylamine; 2 were also reported for arylcyclohexylamines.

### 4.13. NPS Cardiotoxicity Implications

The exact incidence of toxic myocardial injury and consequent toxic cardiomyopathy is difficult to establish [87] due to the high degree of variability in toxic exposure and to the presence of any underlying cardiac conditions. Toxic damage could therefore trigger a series of reactions in cardiac cells leading to changes in myocardial morphology, biochemistry and physiology. For these reasons, mechanisms behind the development of toxic cardiomyopathy appeared multifactorial and complex. In fact, the effects of drugs on the heart may be aggravated in the presence of comorbidities/cotreatments since they affect ion channel expression and/or activity, mitochondrial function, electro-mechanical coupling and modification of extracellular matrix composition favoring the induction of arrhythmias, contractile dysfunction and, potentially, cardiomyocyte death.

Some toxic substances [88] are able to produce direct or indirect toxicity to cardiomyocytes. Direct toxicity, present in several experimental models of toxic cardiomyopathy, can be mediated by reactive oxygens species (ROS) production, which determines lipid peroxidation through membrane interactions and mitochondrial oxidative stress, or by Mitogen-Activated Protein Kinases (MAPKs). The generated ROS may also further induce DNA damage, especially DNA single-strand breaks.

Indirect toxicity [89] can instead be mediated by a neurohormonal response involving excessive catecholamine release or blocking noradrenaline-reuptake mechanism. The major representatives are indirect sympathomimetics and nonselective sympathomimetics, which affect both the vascular system (via noradrenaline actions on α1-adrenergic receptors) and the heart (due to noradrenaline actions on β1-adrenoreceptors).

## 5. Conclusions

We undertook a systematic literature review to identify the NPS-related intoxications and fatalities during the COVID-19 pandemic. Although we observed a large diffusion of several NPS from different structural classes, from January 2020 to March 2022, synthetic opioids emerged as the most-abused class of NPSs, causing fatalities when consumed, both alone and in combination with other NPS classes. New designer benzodiazepines were frequently found alongside fentanyl analogues in drug-related deaths displaying respiratory depressant properties. Such combinations may substantiate the assumption that, in times of stress and crisis, vulnerable people and polydrug users prefer drugs that can be used in solitude to escape anxiety, uncertainty and distress—e.g., that generated by the COVID-19 pandemic. Moreover, we observed that many intoxications due to single NPS-exposure were caused by synthetic cannabinoids, followed by synthetic cathinones and synthetic opioids. Considering intoxications observed in polydrug users, a mixture of different NPS classes were observed, including synthetic cathinones, synthetic opioids, arylcyclohexylamines, natural NPS and designer benzodiazepines.

From 2020 to 2022 the number of NPS exposures changed, from 123 cases in 2020 to 71 cases in 2021 and 21 cases up from January–March 2022. Nonetheless, this represented a decrease during those years, though we were unable to clarify whether the cause of this decline was caused by a delay in toxicological processing or a decrease in drug consumption.

The wide range of NPS chemical structures represents the main challenge for their detection in toxicological laboratories. Reference standard availability, method development, the scarcity of knowledge about NPS metabolism and the serious ongoing health emergency could have caused a loss of information about NPS-related intoxications and fatalities. Therefore, hyphenated analytical techniques, validated and updated methods are necessary to ensure reliable results in the identification of analytes. In acknowledgment of this concern, we observed that liquid chromatography mass spectrometry was the most suitable technique used in the cases reported [90,91].

The main limitations of this study were (1) the lack of information regarding concentrations in biological matrices of NPS and their metabolites and (2) the difficulty in identifying which NPS class caused intoxication or death. Comprehension of the analytical data is needed to better understand NPS toxic/lethal blood concentrations, metabolism and postmortem redistribution.

## Figures and Tables

**Figure 1 biology-12-00273-f001:**
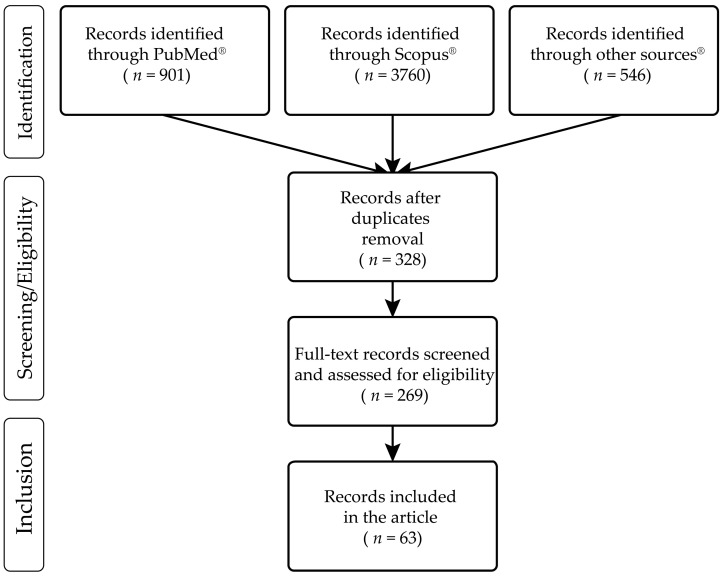
Prisma flowchart of the literature search on NPS intoxications and fatalities cases during the pandemic period (January 2020–March 2022).

**Figure 2 biology-12-00273-f002:**
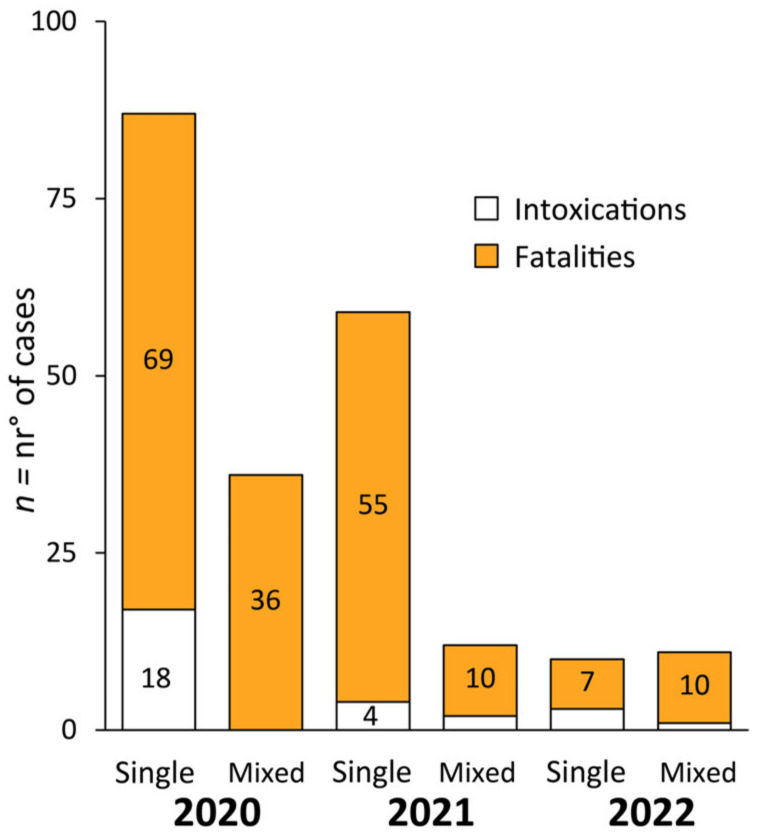
Number of single- and mixed-NPS intoxications and fatalities during the pandemic period by year (January 2020–March 2022).

**Figure 3 biology-12-00273-f003:**
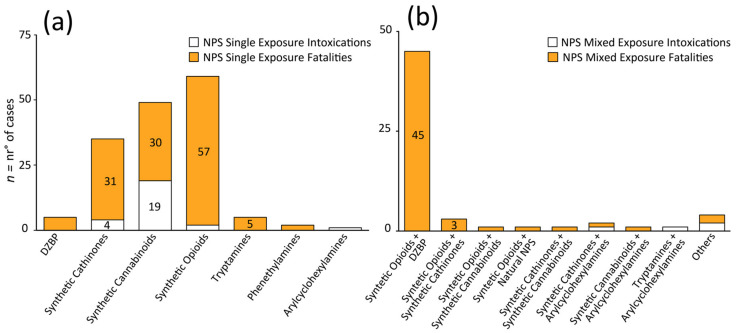
Number of NPS single intoxications and fatalities. (**a**) NPS mixed intoxications and fatalities (**b**) during pandemic period (January 2020–March 2022). Designer Benzodiazepines, DZBP.

**Figure 4 biology-12-00273-f004:**
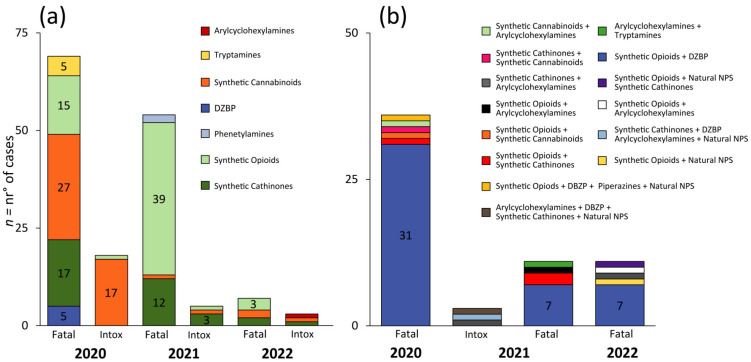
(**a**) Number of NPS single intoxications and fatalities cases; (**b**) NPS mixed intoxications and fatalities cases by year (January 2020–March 2022). Designer Benzodiazepines, DBZP; others (see Section 3.3 and Section 3.4).

## Data Availability

Not applicable.

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
