# Peer review of "New Psychoactive Substances Intoxications and Fatalities during the COVID-19 Epidemic"

_biology, 2023, doi:10.3390/biology12020273_

Round 1

Reviewer 1 Report

Dear authors, I greatly appreciated the theme of your work.

Emerging drugs pose a challenge to forensic toxicology. The toxicological analysis on the "white deaths" are often negative, because it is not known which metabolite to look for.

The article has strong limitations, does not follow the journal's guidelines, and does not conduct a proper review of the literature.

As for the reproducibility of the experiment, entering the indicated search terms in pubmed database, there are no results.

best regards

Author Response

Dear Reviewer,

We are grateful for the time and effort you took reviewing the manuscript; here are the answers to your suggestions and constructive criticisms:

Dear authors, I greatly appreciated the theme of your work.

  1. Emerging drugs pose a challenge to forensic toxicology. The toxicological analysis on the "white deaths" are often negative, because it is not known which metabolite to look for.

In agreement with the reviewer's comment, we stress the problematic New Psychoactive Substances (NPS) worldwide issue that is due to lack of data, reference standard availability and knowledge of NPS metabolism.

  1. The article has strong limitations, does not follow the journal's guidelines, and does not conduct a proper review of the literature.

As suggested, we improved the manuscript adding graphical abstract, Prisma flowchart (Figure 1), and the Institutional Review Board Statement, Informed Consent Statement and Data Availability Statement were not required as the article was based entirely on published data (Line 498-500, page 27).

  1. As for the reproducibility of the experiment, entering the indicated search terms in PubMed database, there are no results.

As suggested, we modified the statement to better clarify the terms used in PubMed for our literature search. PubMed and the other engines searched did not allow “..” in the query search line. Thus, alone and in combination refers to “OR” and “AND”, respectively. (Line 111-120, page 3)

Thanks again for your valuable contribution to the review process.

Devoutly,

Prof. Simona Zaami and coauthors

Reviewer 2 Report

This paper has the potential to be a useful and timely update regarding the epidemiology of NPS during COVID-19. Clearly, a lot of time and effort has gone into data collection, etc. However, there are major issues in terms of the lack of information regarding methodology which would allow replication, and also in respect of what information is presented that would allow epidemiological interpretation. The limitations of the information presented are not dealt with adequately.

Happy to look at a revised version.

Please see attached annotated pdf for all comments and suggestions

Author Response

Dear Reviewer,

We are grateful for the time and effort you took reviewing the manuscript; here are the answers to your suggestions and constructive criticisms:

This paper has the potential to be a useful and timely update regarding the epidemiology of NPS during COVID-19. Clearly, a lot of time and effort has gone into data collection, etc. However, there are major issues in terms of the lack of information regarding methodology which would allow replication, and also in respect of what information is presented that would allow epidemiological interpretation. The limitations of the information presented are not dealt with adequately.

Happy to look at a revised version. 

Please see attached annotated pdf for all comments and suggestions.

Comments

  1. Add s to “Substances” or change start of the title to “Intoxications and Fatalities from New Psychoactives Substances during….”

As suggested, the letter s was added to Psychoactives.

  1. Line 22 Insert “literature” after “systematic”

As suggested, literature was added after systematic (Line 23, page 1)

  1. Line 22 Explain “NPS” here

As suggested, NPS was defined (Line 23, page 1)

  1. Line 27 See earlier comment about NPS

As suggested, this was amended (Line 28, page 1)

  1. Where were these reported?

As suggested, country of occurrence was added to the cases when reported (Line 28-29, page 1)

  1. Replace “underestimated” with an under-estimation

As suggested, under-estimation replaced “underestimated”. (Line 33, page 1)

  1. Insert comma after “concern”

As suggested, a comma was added after “concern”. (Line 35, page 1)

  1. Insert “systematic” before “literature”

As suggested, “systematic” was inserted before “literature”. (Line 37, page 1)

  1. Insert “as” after “such”

As suggested, “as” was inserted after “such”. (Line 38, page 1)

  1. What were the search terms when the searches done?

As suggested, we added the search terms in the abstract (Line 40-45, page 1)

  1. Where were they reported i.e., countries?

As suggested, we added the Country/State for the cases when reported (Line 46, page 1)

  1. Did Covid cause delays in processing toxicological, autopsy and death investigation?

As suggested, we modified the sentence clarifying the statement. (Line 54-57 page 2)

  1. What is the relevance of SARS?

As suggested, we removed the keywords “Sars-Cov19” that was redundant. (Line 58, page 2)

  1. Evidence for this claim?

As suggested, we added 2 references to justify this claim. (Line 70, page 2)

  1. Explain this reduced capability

As suggested, we modified the sentence, and added a reference. (Line 95-97, page 2)

  1. Very important to state when the searches were done!

As suggested, we indicated in the text that the literature search was done in March 2022. (Line 107, page 3)

  1. Did you think to search Google Scholar to identify any conference abstracts or master’s or doctoral theses?

For the literature search we focused on the full-text articles published in international journals instead of preliminary data presented as conference abstracts, which are comprised of shorter sentences and succinct text presenting only the most important findings. Master’s or doctoral theses are usually published as peer-reviewed full-text articles.

  1. Are there any published statistics which can help provide context for an examination of these case-series/case-reports?

See, for example, https://www.ons.gov.uk/peoplepopulationandcommunity/birthsdeathsandmarriages/deaths/datasets/deathsrelatedtodrugpoisoningbyselectedsubstances

Thank you for the suggestion, but unfortunately, to our knowledge, there are no published statistics for this topic as the one suggested.

  1. Clarify I any ‘wildcards’ were us

As suggested, we re-wrote the search terms for clarification, but we did not fully understand the reviewer’s comment.

  1. Why only these languages?

For the literature search we focused on full-text articles in English, Italian, or French language according to the language skills of the authors of our manuscript. Moreover, according to Thomson Reuters®, only 6% of researchers worldwide publish articles in Spanish, less than 1% in Arabic, compared to 95% in English. Furthermore, most reviews limit articles to those written only in English.

  1. Of only biological tissues and/or environmental samples? As suggested, we clarified that only biological samples were reported. (Line 132-133, page 3)

  1. I would expect to see a prima flowchart somewhere

As suggested, we added a Prisma flowchart as figure 1.

  1. How were data extracted, curated and analysed? What software(s) were used? Did more than one individual extract data and or compare data to avoid researcher/selection bias?

We clarified that data were extracted, curated and analysed without software; all reports were manually screened and checked by three of the authors. (Line 134-136, page 3)

  1. Is “patient” the correct term?

As suggested, we corrected the text. (Line 143, page 4)

  1. Any information on place of death e.g., Hospital, which might indicate if interventions had occurred?

As suggested, we added the location of death if it was provided.

  1. Reporting does not necessarily equate to occurrence. You need to give month and year of occurrence

As suggested, we added the month of occurrence if provided in the article. We also added into Table 1 the year and country of occurrence.

  1. You cannot talk about prevalence, you do not have a denominator.

As suggested, we modified the term ‘prevalence’ to ‘percentage’ (Line 149, page 4).

For percentage, the numerator was the NPS exposures that occurred in 2020-2022 and the denominator the total number of cases reported, multiplied by 100.

  1. Where do you discuss this apparent decline? Could this results from a delay of processing toxicology, autopsy and death registration?

We don’t know whether this decline was caused by a delay in toxicological processing or a decrease in drug consumption because it was not clear from the reports. As suggested, we stressed this concept in the text this concept for future consideration (Line 470-472 page 26).

  1. What about mean, range?

Unfortunately, concentration data for the synthetic cannabinoids were not available.

  1. Replace “prevalence” with “common”

As suggested, “prevalence” was substituted with “common” (Line 160; Line 165, page 4).

  1. Pandemic does not need to have initial Capital letter

As suggested, the correction was made. (Line 190, page 6)

  1. Rephrase “until March 2022”, e.g., from “January to March”

As suggested, the correction was made.  (Line 193, page 6)

  1. Replace “prevalent” with “common”

As suggested, the correction was made. (Line 205, page 7)

  1. mean, range?

Unfortunately, concentration data for the synthetic cannabinoids were not available. See earlier comment about using “prevalent”

As suggested, “prevalence” was substituted with “common” (Line 213, page 7)

  1. add ‘s’ to “DBZP”

As suggested, the correction was made. (Line 222, page 7)

  1. Table 1 I would have Table 1 as Supplementary Information

We respectfully disagree with the reviewer, the content of the table, summing data not explained in the manuscript, providing a useful tool for the readers for better understanding of results.

  1. What about NPS in combination with other substances?

The focus of this review is only about NPS consumed alone or in combination. Table 1 provides the other substances detected in the biological matrices.

  1. Reporting between Jan 2020 and March 2022 does not necessarily equate to occurrence during this period. Can you make clear when intoxications and deaths actually occurred. Thus, is very important from an epidemiological perspective

We agree with the reviewer, year and country where the cases occurred are displayed in Table 1 but may not necessarily match with the year of publication of the articles. In some cases, the authors do not provide the exact year and country where the case occurred.

  1. Facilitates what to cross this barrier

As suggested, we rephrased the sentence (Line 295, page 23)

  1. add ‘s’ to “DBZP”  

As suggested, the correction was made. (Line 309, page 23)

  1. Many of the comment about quantitate of precursors and drug being seized appear to be gratuitous, and do not really mesh in/integrate with the commentary

As suggested, we removed the statement (Line 318-319 page 23)

  1. “class” should be “classes”

As suggested, the correction was made. (Line 337, page 23)

  1. “have” should be “has”

As suggested, the sentence was revised. (Line 352, page 24)

  1. add ‘s’ to “Cannabinoids”

As suggested, the correction was made. (Line 359, page 24)

  1. Replace ‘its’ with “cannabis”

As suggested, the correction was made. (Line 363, page 24)

  1. Evidence for this statement?

We apologize for the error and revised the reference.  (Line 372, page 24)

  1. This sentence appears to be a gratuitous insertion

As suggested, we removed the statement. (Line 406-407, page 25)

  1. What is a “toxic insults”? I have never come across this term in 35 years of research on drugs

As suggested, we revised the sentence. (Line 435, page 25)

  1. Replace “emphasized” with “increased” or “aggravated” or “enhanced”

As suggested, we revised the sentence. (Line 439, page 26)

  1. Explain “ROS” stands for

As suggested, we defined “ROS”. (Line 445, page 26)

  1. The discussion section need to reflect on the comprehensiveness (or otherwise) of the cases presented. Nothing mentioned about inherent danger of researcher/selection bias

All reports included in the review were carefully screened by three of the authors. Inclusion criteria were defined prior to the research. Moreover, a double check was performed by the authors to avoid the risk of researcher/selection bias. No tools were used to assess this risk. We clarify this aspect in Materials and Method section (Line 134-136, page 3)

  1. It is unclear whether pattern in numbers of intoxications and particularly deaths, could be as a direct impact of Covid-19 on the processing of toxicological, autopsy and death investigations – due to lack of staff, reference sample samples, availability of expert and other witnesses at inquests, coroners/MEs’, investigations, etc.

As the World Health Organization (WHO) declared a global pandemic due to SARS-CoV-2, control measures impacted health care services at limited capacity worldwide. Therefore, we cannot rule out a direct impact of Covid-19 also on the processing of toxicological, autopsy and death investigations, although none of the full-text included in our study detailed this important aspect.

  1. There is no mention of how Covid-19 may have influenced cause of deaths, possible interventions, etc

This is the same comment mentioned just above. From the EMCDDA report “Impact of COVID-19 on drug markets, use, harms and drug services in the community and prisons,” this aspect of pandemic effects is still unknown. It also might be that forensic and toxicological resources were reassigned to handle post-mortem investigations as well.

  1. the comments in the final paragraph of the Conclusion appear from nowhere; they need to mentioned in the Discussion

According to the reviewer we may not be enough clear, indeed, we modified the last paragraph for a better comprehension of the statement. (Line 311-313, page 23).

  1. Any ethical considerations? Even if not needed, you need to explain why….

As suggested, the Institutional Review Board Statement, Informed Consent Statement and Data Availability Statement were added (Line 498-500 page 27).

Thanks again for your valuable contribution to the review process.

Devoutly,

Reviewer 3 Report

Lo Faro et al. present a comprehensive, systematic review of all published NPS-related 22 intoxications and fatalities during the COVID-19 pandemic (from January 2020 to March 2022). The work is well organized and comprehensively described but it is my opinion that it is not focused on the Special Issue items (“determination of biomarkers of exposure to psychoactive substances” and "...analytical challenges posed by the characterization and determination of the most effective biomarkers of traditional and new psychoactive substances, within the framework of clinical and forensic toxicology").

The paper should be improved with the description/comparison of used analytical methods and  with the comparison of the reported data of intoxications and fatalities with the same published before the COVID-19 pandemic (same time range) to better understand the effects of the pandemic on drug consumptions.

Author Response

Dear Reviewer,

We are grateful for the time and effort you took reviewing the manuscript; here are the answers to your suggestions and constructive criticisms:

  1. The paper should be improved with the description/comparison of used analytical methods and  with the comparison of the reported data of intoxications and fatalities with the same published before the COVID-19 pandemic (same time range) to better understand the effects of the pandemic on drug consumptions.

As suggested, we added a paragraph in the result section and conclusion about the analytical methods. (Line 264-271, page 8; Line 474-480, page 26). Moreover, we clarify our aims in the introduction, to better understand our focus. (Line 99-102, page 3)

We thank the Reviewer for her/his suggestion. In our opinion, the Reviewer highlighted an interesting question, which requires an in depth and comprehensive analysis of the literature. As declared, we focused on NPS-related intoxications and fatalities during the COVID-19 pandemic (from January 2020 to March 2022), and would require further expansion of our analysis, beyond the initial scope of the study.

Thanks again for your valuable contribution to the review process.

Devoutly,

Prof. Simona Zaami and coauthors

Reviewer 4 Report

New Psychoactive Substance Intoxications and Fatalities during the COVID-19 Epidemic

The purpose of this review is to analyze and report all published NPS-related intoxications and fatalities during the COVID-19 pandemic (from January 2020 to March 2022). During this period, synthetic cannabinoids followed by synthetic cathinones and synthetic opioids were the cause of single NPS-exposure.

Comments

The manuscript is very interesting and topical. There are no similar publications in the literature. With the Covid 19 pandemic, drug use trends have changed so the study conducted by the authors provide an interesting overview of drug use during this period. English language and style are fine.

Only small improvements are recommended as follows:

·      The format of all references should follow the style request of the journal;

·      Check the surname and e-mail of the author “Berretta P”.

The article can be accepted after the required revisions have been made.

Author Response

Dear Reviewer,

We are grateful for the time and effort you took reviewing the manuscript; we have attempted to improve the manuscript and the reference pool.

Thanks again for your valuable contribution to the review process.

Devoutly,

Round 2

Reviewer 2 Report

The authors have taken on board on all the comments and dealt with them in a constructive way. This has helped improve this useful contribution to the literature.

Author Response

Dear Reviewer,

My coauthors and I would like to thank you for your important contribution to the review process and insightful suggestions enabling us to improve the article.

Best regards,

Prof. Simona Zaami and coauthors

Reviewer 3 Report

In my previous review, I suggested to improved the manuscript with "the comparison of the reported data of intoxications and fatalities with the same published before the COVID-19 pandemic (same time range) to better understand the effects of the pandemic on drug consumptions." because the title hints that the pandemic affected the events described in the review. It was not my intent to ask the Authors to repeat the same in depth and comprehensive analysis of the literature but on another period but to explain the influence of the pandemic period on the analysis made.

Author Response

Dear Reviewer,

My coauthors and I are grateful for your valuable contribution to the review process.

Best regards,

Prof. Simona Zaami and coauthors